# Genetic dissection of the RNA polymerase II transcription cycle

Shao-Pei Chou[1], Adriana K Alexander[1,2], Edward J Rice[1], Lauren A Choate[1], Charles G Danko[1,2]*

[1]Baker Institute for Animal Health, College of Veterinary Medicine, Cornell University, Ithaca, United States; [2]Department of Biomedical Sciences, College of Veterinary Medicine, Cornell University, Ithaca, United States

**Abstract** How DNA sequence affects the dynamics and position of RNA Polymerase II (Pol II) during transcription remains poorly understood. Here, we used naturally occurring genetic variation in F1 hybrid mice to explore how DNA sequence differences affect the genome-wide distribution of Pol II. We measured the position and orientation of Pol II in eight organs collected from heterozygous F1 hybrid mice using ChRO-seq. Our data revealed a strong genetic basis for the precise coordinates of transcription initiation and promoter proximal pause, allowing us to redefine molecular models of core transcriptional processes. Our results implicate DNA sequence, including both known and novel DNA sequence motifs, as key determinants of the position of Pol II initiation and pause. We report evidence that initiation site selection follows a stochastic process similar to Brownian motion along the DNA template. We found widespread differences in the position of transcription termination, which impact the primary structure and stability of mature mRNA. Finally, we report evidence that allelic changes in transcription often affect mRNA and ncRNA expression across broad genomic domains. Collectively, we reveal how DNA sequences shape core transcriptional processes at single nucleotide resolution in mammals.

*For correspondence:
dankoc@gmail.com

Competing interest: The authors declare that no competing interests exist.

## Editor's evaluation

This study exploits naturally occurring single nucleotide polymorphisms between mouse strains to provide novel information on how DNA sequence affects RNA polymerase II transcription initiation, termination, and pausing. The strength of this study lies in assessing how naturally occurring allele-specific polymorphisms impact the various steps in the transcription cycle in an unbiased and genome-wide manner.

## Introduction

Transcription by RNA polymerase II (Pol II) results in the synthesis of mRNAs encoding all protein-coding genes. Pol II transcription is a cyclic process that can be divided into stages representing transcription initiation, pause, elongation, and termination (*Fuda et al., 2009*; *Jonkers and Lis, 2015*). During the first stage of the transcription cycle, RNA polymerase is initiated on regions of accessible chromatin by the collective actions of transcription factors and co-factors that recruit the pre-initiation complex (PIC), melt DNA, and initiate Pol II (*Grünberg et al., 2012*; *Haberle and Stark, 2018*; *Murakami et al., 2013*; *Tsai and Sigler, 2000*). After initiation, Pol II pauses near the transcription start site of nearly all genes in metazoan genomes (*Jonkers et al., 2014*; *Muse et al., 2007*; *Rougvie and Lis, 1988*). Pol II is released from pause by transcription factors and a key protein kinase complex (P-TEFb), a tightly regulated step that controls the rates of mRNA production (*Danko et al., 2013*; *Mahat et al., 2016a*; *Rahl et al., 2010*; *Zeitlinger et al., 2007*). After pause release, Pol II elongates through gene

bodies, which in some cases cover more than 1 MB of DNA in mammals (*Carninci et al., 2005*). Finally, the co-transcriptionally processed pre-mRNA is cleaved from the elongating Pol II complex, allowing termination and recycling of Pol II (*Cho et al., 1999*; *O'Sullivan et al., 2004*; *Rosonina et al., 2006*). Intensive research efforts during the past 30 years have provided detailed knowledge of the proteins, RNAs, and other macromolecules that facilitate Pol II transcription (*Gilchrist et al., 2010*; *Miller et al., 2001*; *Nechaev et al., 2010*; *Orphanides et al., 1998*; *Ranish et al., 1999*).

We still have a relatively rudimentary understanding about how DNA sequence influences each step in the Pol II transcription cycle. Of all stages in Pol II transcription, the DNA sequence determinants of transcription initiation are perhaps the best characterized. DNA sequence motifs such as the TATA box, B recognition element (BRE), and the initiator motif are reported to bind proteins in the PIC and thereby set the transcription start site (*Carninci et al., 2006*; *Nilson et al., 2017*; *Smale and Baltimore, 1989*). SNPs affecting these core transcription initiation motifs can affect the rates of mRNA production indicating the central importance of DNA sequence affecting transcription initiation (*Kristjánsdóttir et al., 2020*). In addition, DNA sequence motifs that correlate with Pol II pausing and termination have also been reported (*Gressel et al., 2017*; *Schwalb et al., 2016*; *Tome et al., 2018*; *Watts et al., 2019*). In all cases, these studies show that DNA sequence motifs involved in Pol II transcription are weak, degenerate, and spread across wide genomic regions. Therefore, we are still a long way from developing predictive models that describe interactions between DNA sequence context and the steps and dynamics of Pol II transcription.

Here, we used naturally occurring genetic variation between heterozygous F1 hybrid mice to understand how DNA sequence differences between alleles affect the Pol II transcription cycle. We generated an atlas of the position and orientation of RNA polymerase II in eight organs collected from three primary germ layers using ChRO-seq (*Chu et al., 2018*). Our detailed analysis reveals insight into how DNA sequence shapes each of the stages during the Pol II transcription cycle. For initiation, our results support a new model in which Pol II selects initiation sites through a process similar to Brownian motion. We show that Pol II pauses on a C base following a G-rich stretch and is affected by short indels between the initiation and pause position, together explaining more than half of the observed changes in Pol II pause position. Finally, we reveal substantial allelic differences in the position of transcription termination, which in some cases affect the primary structure and transcript stability of the mature mRNA. Collectively, we provide new insight into how DNA sequence shapes core transcriptional processes in mammals.

## Results

### Atlas of allele-specific transcription in F1 hybrid murine organs

We obtained reciprocal F1 hybrids from two heterozygous mouse strains, C57BL/6 (B6) and Castaneus (CAST) (*Figure 1A*). Mice were harvested in the morning of postnatal day 22–25 from seven independent crosses (3 x C57BL/6 x CAST and 4 x CAST x C57BL/6; all males). We measured the position and orientation of RNA polymerase genome-wide in eight organs using a ChRO-seq protocol designed to improve the accuracy of allelic mapping by extending the length of reads using strategies similar to length extension ChRO-seq (*Chu et al., 2018*) (see Materials and methods). We obtained 376 million uniquely mapped ChRO-seq reads across all eight organs (21–86 million reads per organ; *Supplementary file 1*) after sequencing, filtering, and mapping short reads to individual B6 and CAST genomes. Hierarchical clustering using Spearman's rank correlation of ChRO-seq reads in GENCODE annotated gene bodies (v.M25) grouped samples from the same organ (*Figure 1B*), as expected based on studies comparing multiple organs across different species (*Gilad and Mizrahi-Man, 2015*; *Merkin et al., 2012*). Additionally, organs with similarities in organ function clustered together, for instance: heart and skeletal muscle, and large intestine and stomach (*Figure 1B*).

We identified 1374 genes and lincRNAs with strong evidence that transcription was significantly higher across the gene body on either the B6 or CAST allele in at least one organ, comprising about 8% of the 17,703 annotated genes. Visualizing the pattern of allelic transcription revealed that most allelic changes were specific to a single organ (*Figure 1C*). A minority of annotated genes showed evidence of genomic imprinting (n=51), and these were generally imprinted in all organs analyzed here (*Figure 1C*). Organ-specific allelic biased transcription was not explained by either false-negatives in putatively unbiased organs (*Figure 1—figure supplement 1* A,B) or by organ-specific

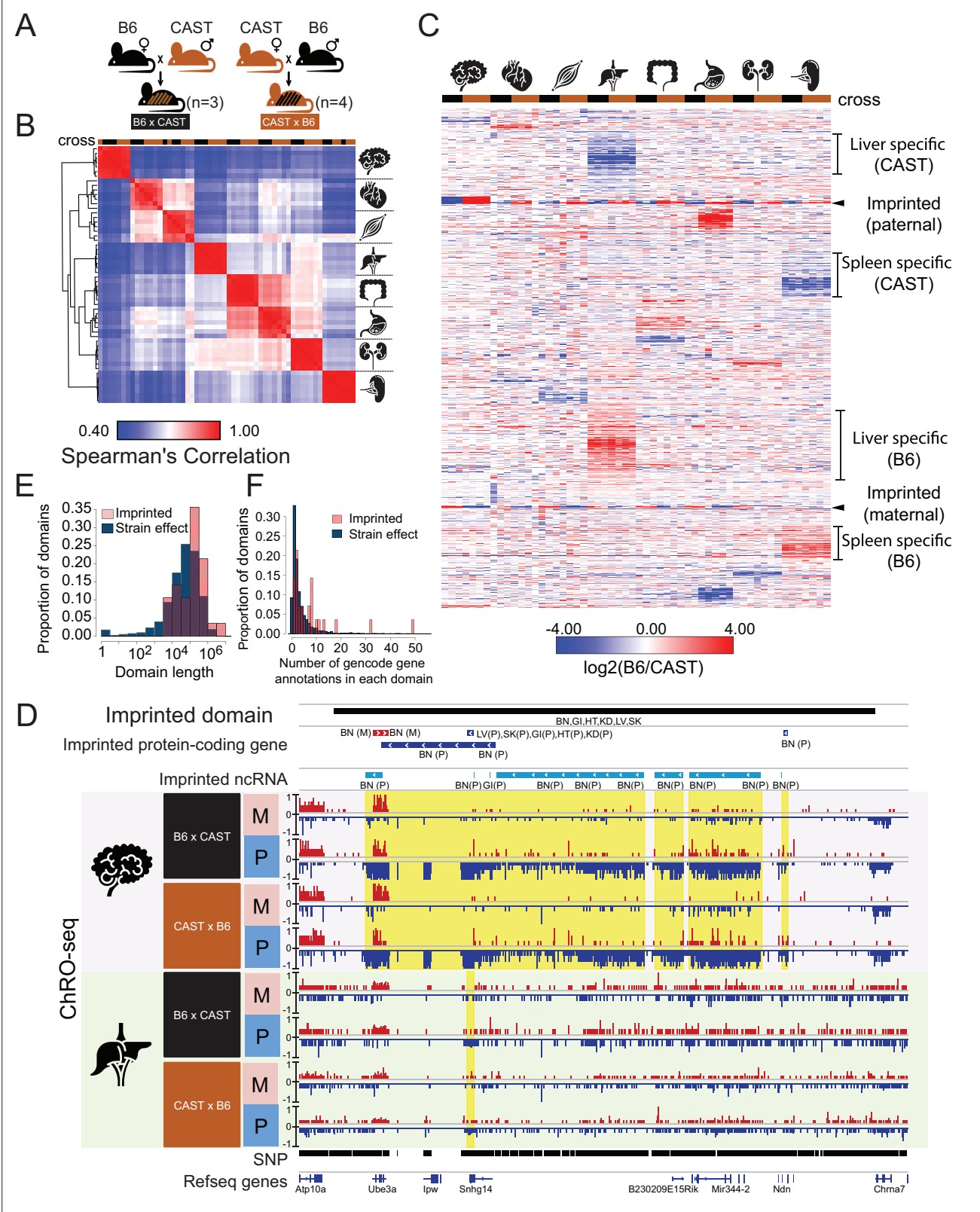

**Figure 1.** Reciprocal hybrid cross to understand the Pol II transcription cycle. (**A**) Cartoon illustrates the reciprocal F1 hybrid cross design between the strains C57BL/6 J (**B6**) and CAST/EiJ (CAST). We have seven independent crosses (3 x C57BL/6 x Cast and 4 x Cast x C57BL/6). (**B**) Spearman's rank correlation of ChRO-seq signals in gene bodies. The color on the top indicates the direction of crosses: Black is B6 x CAST, brown is CAST x B6. The cartoon on the right indicates the organ each sample was harvested from. (**C**) Heatmap shows the allelic bias (log2[B6 /CAST]) in GENCODE (vM25)

*Figure 1 continued on next page*

**Figure 1 continued**

annotated genes >10 kb in size. Row order is determined by hierarchical clustering (using 1 - Pearson correlation); Column order was set manually. Several of the different gene cluster interpretations are shown by the writing on the right. (**D**) The browsershot shows an example of ChRO-seq data that has an imprinted domain (top row). The second and third rows show the imprinted protein-coding genes and imprinted non-coding RNA (ncRNA) from all organs. (BN:brain, LV:liver, SK:skeletal muscle, GI: large intestine, HT: heart, KD: kidney, P: paternal, M: maternal). The yellow shade indicates the imprinted regions in the brain and liver. (**E**) The histogram shows the domain length as a function of the proportion of domains. (**F**) The histogram shows the number of GENCODE (vM25) gene annotations in each domain.

The online version of this article includes the following figure supplement(s) for figure 1:

**Figure supplement 1.** Validation of organ specific allelic biased domains.

differences in gene expression (*Figure 1—figure supplement 1C*). Notably, even organs with similar gene expression patterns (e.g. heart and skeletal muscle; large intestine and stomach) did not show much correlation in allelic differences between B6 and CAST. Taken together, these results show that most allelic differences in gene transcription were organ specific and showed heritability patterns that were consistent with a genetic cause.

Examination of genome browser tracks revealed that genes with allele-specific transcription frequently clustered in genomic regions that had multiple separate transcripts with allelic differences. For instance, the imprinted domain associated with Angelman syndrome contained twenty ncRNAs and four genes that were transcribed more highly from the paternal allele and two genes transcribed from the maternal allele (*Figure 1D*). Multiple transcriptional changes occurring across a single locus in this example, and others like it that fit both imprinting and genetic inheritance patterns (*Figure 1—figure supplement 1D*), are consistent with differences affecting regulatory regions that control the activity of broad transcription domains. We asked whether genes located near one another shared allelic changes more frequently than expected if changes were caused by independent genetic or epigenetic differences. Indeed, we found that genes with allelic bias were significantly more likely to have allelic changes in the expression of an adjacent transcript compared with genes which were not changed (brain: p-value = 5.35e-4, liver: p-value < 2.2e-16; Fisher's Exact Test).

To identify domains with evidence of allelic bias in an annotation-independent manner, we next analyzed ChRO-seq data from all eight organs using AlleleHMM (*Chou and Danko, 2019*). We identified 3494 domains that showed consistent evidence of allelic imbalance (*Figure 1—figure supplement 1E*, *Supplementary file 2*). The majority of these domains (n=3466) had consistent effects in each mouse strain (called strain-effect domains), the pattern expected if allelic imbalance was caused by DNA sequence differences between strains. Twenty-eight domains showed consistent evidence of genomic imprinting (imprinted domains). On average, both strain-effect and imprinted domains spanned broad genomic regions (~10–1000 kb; *Figure 1E*) that were frequently composed of two or more transcription units, including annotated genes and ncRNAs (either long intergenic ncRNAs and/ or enhancer-templated RNAs; *Figure 1F*). Imprinted domains tended to be larger, have allelic differences detected across multiple tissues, and affected larger numbers of genes than strain effect domains (*Figure 1E–F*). However, despite the overall trend toward larger imprinted domains, we nevertheless did identify many cases of larger domains containing allele-specific differences.

We conclude that allelic differences frequently alter regulatory processes that impact multiple transcription units across a locus. Potential mechanisms may include non-coding RNAs that act in a manner analogous to Xist in X-chromosome inactivation, differentially methylated regions involved in imprinting, or enhancers that regulate the expression of multiple transcripts simultaneously (*Delaneau et al., 2019*; *Kumasaka et al., 2019*; *Rennie et al., 2018*).

## Widespread genetic changes in transcription initiation

Having validated our experimental dataset and examined the broad patterns of allelic differences across organs, we next set out to dissect the genetic basis for each stage of the Pol II life cycle. We first focused on defining allele-specific patterns of transcription initiation. The 5' end of nascent RNA, denoted by the 5' end of paired-end ChRO-seq reads, marks the transcription start site (TSS) of that nascent RNA (*Kwak et al., 2013*; *Tome et al., 2018*). We identified TSSs in which at least 5 unique reads share the same 5' end inside of regions enriched for transcription initiation identified using dREG after merging all samples from the same tissue (*Wang et al., 2019*). Using a recently described hierarchical strategy (*Tome et al., 2018*), we grouped candidate TSSs into transcription start clusters

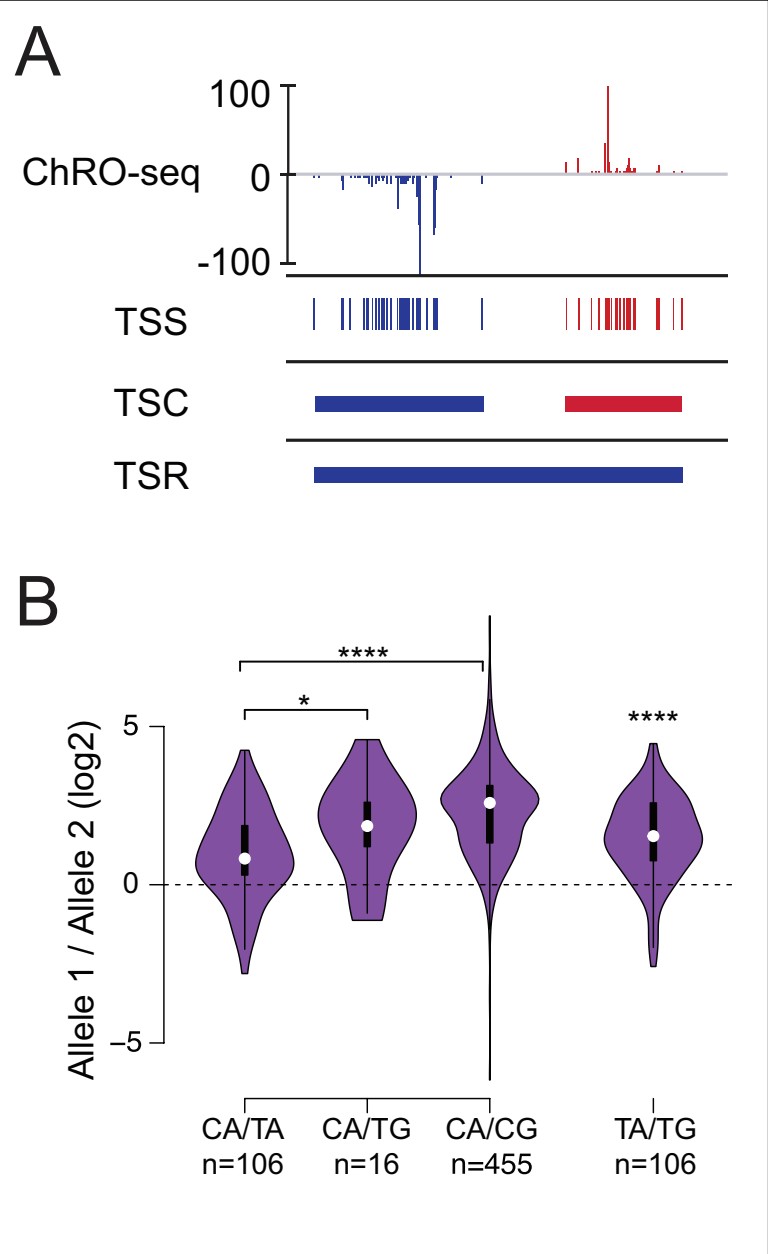

**Figure 2.** Allelic changes in TSSs reveal a hierarchy of initiator dinucleotides. (**A**) The ChRO-seq signals of the 5' end of the nascent RNA were used to define transcription start sites (TSSs), or the individual bases with evidence of transcription initiation. TSSs within 60 bp were grouped into transcription start clusters (TSCs). The broad location of transcription start regions (TSRs), which comprise multiple TSCs, was determined using dREG. (**B**) Violin plots show the ratio of allelic bias between the dinucleotides indicated on the X axis. Asterisk denote statistical significance (* <0.05, ** p<0.01, *** p<0.001, **** p<0.0001) using a two-sided Wilcoxon rank sum test.

The online version of this article includes the following figure supplement(s) for figure 2:

**Figure supplement 1.** Validation of TSS identification using paired-end ChRO-seq data.

(TSCs) and defined the maximal TSS as the position with the maximum 5' signal within each TSC (*Figure 2A*; *mid*). TSCs were grouped in turn into transcription start regions (TSRs), that are composed of multiple nearby TSCs based on the boundaries established by dREG (*Figure 2A*; *bottom*).

To verify that our pipeline identified TSSs accurately, even in the absence of enzymatic enrichment for capped RNAs, we examined whether candidate TSSs were enriched for the initiator DNA sequence element, a defining feature of Pol II initiation (*Kaufmann and Smale, 1994*; *Smale and*

*Baltimore, 1989*). Most organs, especially brain and liver (which were the most deeply sequenced and are the focus of the analysis below unless otherwise specified), had high information content showing the initiator element at maxTSSs (*Figure 2—figure supplement 1A*). Moreover, the relationship between read depth and TSSs was similar to those recently reported in human cells (*Tome et al., 2018*; *Figure 2—figure supplement 1B*). Finally, in K562 cells our pipeline identified TSSs that were supported by PRO-cap data better than existing human gene annotations (*Figure 2—figure supplement 1C*).

Allelic changes in TSSs were common, occurring in ~16–34% of TSCs tagged with SNPs (n=1,109–5,793; binomial test FDR <0.1; *Supplementary file 3*). Changes in TSSs were highly enriched within strain effect domains identified by AlleleHMM (*Chou and Danko, 2019*) (odds ratios 3.5–5.4; p<2.2e-16, Fisher's exact test). Therefore, many of these allelic changes in TSSs likely reflect allelic changes in the rates of transcription initiation on the gene secondary to allelic differences in transcription factor binding or other regulatory processes. These mechanisms have been explored extensively elsewhere (*Battle et al., 2014*; *Chen et al., 2016*; *Lappalainen et al., 2013*; *Montgomery et al., 2010*; *Pickrell et al., 2010*). Throughout the remainder of this paper, we focus on defining the effects of DNA sequence on core transcriptional processes.

We used genetic differences between alleles to define the relative strength of different initiator dinucleotides. The initiator motif is perhaps the best characterized feature of Pol II initiation and is most commonly characterized by a CA dinucleotide, but the sequence preferences of the initiator motif are weak and other dinucleotides are relatively common (*Smale and Baltimore, 1989*). We examined how changes between initiator dinucleotides affected the abundance of ChRO-seq reads. We identified the set of all max TSSs which had DNA sequence differences between B6 and CAST alleles and measured the magnitude of change in ChRO-seq signal. Our analysis revealed a hierarchy of initiator dinucleotides that impact initiation frequency with different magnitudes (*Figure 2B*). CA initiators which changed to TA had the lowest magnitude of change in ChRO-seq signal, whereas CA dinucleotides that changed to CG had the largest magnitude of change in signal. CA to TG changes were intermediate between CA to TA and CA to CG. The number of examples for CA to TG was much lower than other dinucleotide combinations because it required two DNA sequence changes. Therefore, we also examined TA to TG changes directly. This revealed that changes in initiator sequence from TA to TG were associated with a higher Pol II on the TA allele. Our results suggest a hierarchy of initiation frequency, in which Pol II prefers to initiate at a CA dinucleotide, followed by TA, TG, and finally CG. This hierarchy observed in mammals differs substantially from yeast, which favors CG over either TA or TG (*Zhu et al., 2021*), but is more closely aligned with the importance of A in mammals based on promoter mutagenesis studies (*Javahery et al., 1994*; *Vo Ngoc et al., 2017*; *Smale and Kadonaga, 2003*).

## Allelic changes in the shape of transcription initiation

We next asked how DNA sequences specify which of the multiple independent initiator dinucleotides capture the majority of Pol II initiation events. We reasoned that cases in which Pol II initiation changed from one TSS to a nearby TSS would be highly informative about this initiation code. We therefore developed a statistical approach based on the Kolmogorov-Smirnov test to identify differences in the shape of Pol II initiation, defined as any allelic difference in the TSS distribution within a TSC (see Materials and methods; our approach is similar to *Policastro et al., 2021*). Our approach identified 1006 (brain) and 1389 (liver) TSCs in which the shape of the 5′ end of mapped reads within that TSC changed between alleles (FDR ≤ 0.1; Kolmogorov-Smirnov [KS] test) (see examples in *Figure 3A–B*). We note that changes in the shape of signal within TSCs may not always alter the total abundance of Pol II, as compensatory changes in the rates of Pol II initiation at nearby TSSs may compensate for one another. Consistent with this hypothesis, only ~10–20% of the TSCs identified using this approach were also found inside of strain effect domains, indicating that these changes in shape were fundamentally different from changes in initiation rates and reflected a different underlying biological process. Our results are consistent with reports that small shifts in TSS location can confer robustness to gene expression (*Einarsson et al., 2021*).

As TSCs typically span ~80 bp and are frequently comprised of multiple TSSs (*Carninci et al., 2006*; *Tome et al., 2018*), we divided changes in TSC shape into two classes: cases that were driven predominantly by large changes in the abundance of Pol II at a single TSS position and cases in which multiple

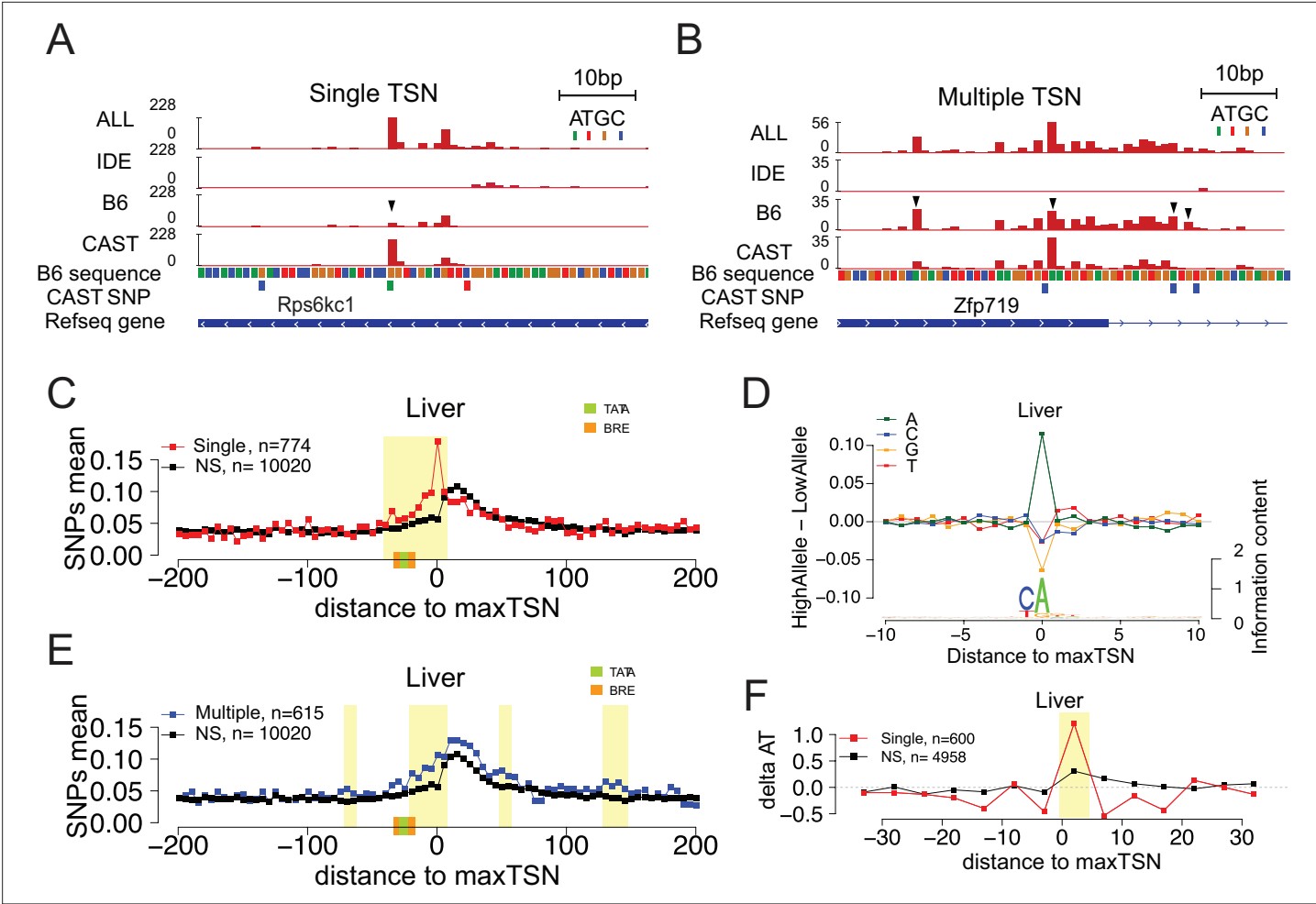

**Figure 3.** Allelic changes in the shape of transcription initiation. (**A**) The browsershot shows an example of allelic differences in the shape of TSC that are predominantly explained by a single TSS position (arrowhead). ALL indicates signal from all reads, IDE indicates signal from reads that are not tagged with a SNP, B6 indicates signal from reads tagged with B6 SNP, CAST indicates signal from reads tagged with CAST SNP. (**B**) The browsershot shows an example of allelic differences in TSC driven by multiple TSSs within the same TSC, arrowheads indicate several prominent positions with allelic differences in Poll abundance. ALL indicates signal from all reads, IDE indicates signal from reads that are not tagged with a SNP, B6 indicates signal from reads tagged with B6 SNP, CAST indicates signal from reads tagged with CAST SNP. (**C**) Scatterplot shows the average SNP counts as a function of distance to the maxTSS at sites showing allelic differences in TSCs driven by a single TSS in the liver. Red denotes changes in TSC shape (Kolmogorov-Smirnov (KS) test; FDR ≤ 0.10); black indicates TSCs without evidence for differences in TSC shape (KS test; FDR >0.90). Dots represent non-overlapping 5 bp bins. Yellow shade indicates statistically significant differences (false discovery rate corrected Fisher's exact on 10 bp bin sizes, FDR ≤ 0.05). Green and orange boxes denote the position of PIC binding motifs. (**D**) The scatterplot shows the average difference in base composition between the allele with high and low TSS use around the maxTSS in single-base driven allele-specific TSCs. The sequence logo on the bottom represents the high allele in single-base driven allele-specific TSCs. The high/low allele were determined by the read depth at maxTSS. (**E**) Scatterplot shows the average SNP counts as a function of distance to the maxTSS at sites showing allelic differences in TSC driven by multiple TSSs in the liver sample. Blue denotes changes in TSC shape classified as multiple TSS driven (Kolmogorov-Smirnov (KS) test; FDR ≤ 0.10); black indicates TSCs without evidence for differences in TSC shape (KS test; FDR >0.90). Dots represent non-overlapping 5 bp bins. Yellow shade indicates statistically significant differences (false discovery rate corrected Fisher's exact on 10 bp bin sizes, FDR ≤ 0.05). (**F**) The scatter plot shows the difference of AT contents between the high and low alleles with the maxTSS and –1 base upstream maxTSS masked. Dots represent 5 bp non-overlapping windows. Red denotes single TSS-driven allele-specific TSCs; black denotes control TSCs with no evidence of allele-specific changes. The yellow shade indicates a significant enrichment of AT (at high allele) to GC (at low allele) SNPs at each bin (size = 5 bp; Fisher's exact test, FDR ≤ 0.05).

The online version of this article includes the following figure supplement(s) for figure 3:

**Figure supplement 1.** Allelic changes in the shape of transcription initiation.

TSSs across the TSC contributed to changes in shape (see Materials and methods). For example, the TSC giving rise to the transcription unit upstream and antisense to *Rps6kc1* had major differences in just one of the TSSs (*Figure 3A*, arrow head), whereas the promoter of *Zfp719* has multiple changes in TSSs within the same TSC (*Figure 3B*, arrow heads). In both examples, allelic changes result in differences in the position of the max TSS between alleles. Each of these two classes comprised approximately half of the changes in TSS shape in all organs (*Figure 3—figure supplement 1A*).

Although the PIC occupies DNA spanning –30 bp to +30 bp relative to the TSS (*Chen et al., 2021*; *He et al., 2016*; *Schilbach et al., 2017*), transcription is also controlled by transcription factors that frequently have factor-dependent positional preferences within the nucleosome free region, located approximately –1 to –110 bp relative to the TSS (*Core et al., 2014*; *Grossman et al., 2018*; *Scruggs et al., 2015*; *Tippens et al., 2020*). To determine which of these sequence elements was most influential for the position of Pol II initiation, we examined the distribution of SNPs centered on allele-specific max TSSs. To control for the ascertainment biases associated with having a tagged SNP in each allelic read, we compared the distribution of SNPs on allele-specific max TSSs with a control set composed of TSSs that have no evidence of allele specificity. Single position driven allele-specific TSCs were associated with a strong, focal enrichment of SNPs around the max TSS (*Figure 3C* [liver] and *Figure 3—figure supplement 1B* [brain]). SNPs within 5 bp of the initiation site explain up to ~15%–20% of single-base driven allele-specific TSCs (*Figure 3E* and *Figure 3—figure supplement 1B*). This was predominantly explained by SNPs at the TSS itself, with an A highly enriched in the allele with highest max TSS usage at that position, consistent with the sequence preference of the initiator motif (*Figure 3D*). By contrast, the enrichment of C in the –1 position of the initiator motif was much weaker than the A at the 0 position. Although this result may partially be explained by a bias in which the C allele does not tag nascent RNA, it was consistent between organs (*Figure 3—figure supplement 1C*) and controlled based on the composition of the background set. Thus, our results suggest that the A in the initiation site may be the most important genetic determinant of transcription initiation, consistent with the Pol II initiation preference analysis conducted above (*Figure 2B*).

Multiple position driven allele-specific TSCs had a weaker, broader enrichment of SNPs (*Figure 3E* [liver] and *Figure 3—figure supplement 1D* [brain], yellow shade indicates FDR ≤ 0.05; Fisher's exact test). Changes in TSC structure driven by multiple, separate TSSs were enriched throughout the ~30 bp upstream, and to some extent downstream, potentially implicating changes in sequence specific transcription factors or other components of the PIC (*Figure 3E* and *Figure 3—figure supplement 1D*). Notably, the enrichment of SNPs within the PIC was also found in single position driven allele-specific TSCs, especially in the window between –30 and +1 relative to the TSS (*Figure 3C*).

We asked whether the TATA box, a canonical core promoter motif that binds TFIID, was also enriched for DNA sequence changes associated with allelic variation in TSS position. As the TATA box only occurs at ~10% of mammalian promoters (*Carninci et al., 2006*; *Lenhard et al., 2012*), we conditioned on the presence of a clear TATA-like motif on at least one of the alleles and asked whether SNPs affecting the TATA box correlated with the magnitude of effect on initiation. We first used a TATA motif that had a general enrichment for AT content, consistent with the degenerate nature of the TATA box in mammals (see Materials and methods). We found that SNPs which changed the TATA motif had a positive correlation with allele specificity, with a slightly stronger correlation in brain than in liver (Pearson's $R$=0.09 [liver], n=201; $R$=0.18 [brain], n=121). The positive correlation was marginally significant in the brain (p=0.04), but not in the liver (p=0.2). Similar results were also obtained with an additional TATA motif that was a stronger match to the classical TATA consensus (TATAAA; p=0.078 [liver], n=218; p=0.051 [brain], n=37). These results are consistent with core promoter motifs playing an important role in the position and magnitude of transcription initiation, but they appear in our analysis to be weaker determinants of the precise initiation site than the initiator motif.

Next, we examined other factors near the TSC, aside from DNA within the initiator motif or other known PIC components, that contributed to the position of TSSs. We noticed a higher frequency of A and T alleles downstream of the initiator motif on the allele with a higher max TSS (*Figure 3D*). We hypothesized that the lower free energy of base pairing in A and T alleles would make them easier to melt during initiation, and could therefore increase the frequency of TSS usage at these positions. Indeed, a more direct examination of AT content in 5 bp windows near the maxTSS identified a significantly higher AT content on the allele with the highest max TSS after masking DNA at positions –1 and 0 to avoid confounding effects of the initiator element

(*Figure 3F* and *Figure 3—figure supplement 1E*). This enrichment of high AT content was consistent in both brain and liver tissue, but was unique to single TSS driven max TSSs (*Figure 3—figure supplement 1F-G*).

We conclude that multiple aspects of DNA sequence, including both sequence motif composition (especially the initiator element) and the energetics of DNA melting, influence TSS choice in mammalian cells.

## Models of stochastic search during transcription initiation

Next, we examined how SNPs that affect a particular TSS impact initiation within the rest of the TSC. In the prevailing model of transcription initiation in *S. cerevisiae*, called the 'shooting gallery model', after DNA is melted, Pol II scans by forward translocation until it identifies a position that is energetically favorable for transcription initiation (*Braberg et al., 2013*; *Giardina and Lis, 1993*; *Kaplan et al., 2012*; *Kuehner and Brow, 2006*; *Qiu et al., 2020*; *Figure 4A*, *left*). Under the shooting gallery model, the transcription start site is determined by the rate of DNA translocation, resulting in a preference for more upstream TSSs, while still retaining the strength of initiator elements. In mammals, Pol II is not believed to scan, but rather each TSS is believed to be controlled by a separate PIC (*Luse et al., 2020*; *Figure 4A*, *right*). We considered how mutations in a strong initiator dinucleotide (CA) would affect transcription initiation under each model. Under the yeast shooting gallery model, we expected CA mutations to shift initiation to the next valid initiator element downstream (*Figure 4A*, *left*). Under the mammalian model, we expected each TSS to be independent and therefore a mutation in the TSS would have no effect on the pattern of nearby initiation sites (*Figure 4A*, *right*).

We analyzed 277 and 372 TSSs in brain and liver, respectively, where the high allele contained a CA dinucleotide while the other allele did not. Candidate initiator motifs within 20 bp of the CA/ non-CA initiation site had slightly more initiation signal on the non-CA allele compared with the CA allele, consistent with the shooting gallery model but inconsistent with the prevailing model of independent TSSs expected in mammals (*Figure 4B*; *purple*). By contrast, TSSs where both alleles contained a CA dinucleotide (n=8147 [brain] and 10,113 [liver]) did not show this same effect (*Figure 4B*; *gray*). The difference was found for both adjacent CA dinucleotides and for weaker candidate (Py)(Pu) initiator elements, and was consistent across both single-base and multiple-base TSC configurations (*Figure 4—figure supplement 1*). These results appear more consistent with the yeast, rather than the current mammalian models, despite the fact that no scanning mechanism has been reported to date in mammals.

Unexpectedly, we also observed consistently higher initiation signals on the non-CA allele both upstream and downstream of the initiator dinucleotide (*Figure 4C*). The signal for an increase on the non-CA allele stretched up to 20 bp both upstream and downstream of the CA/ non-CA dinucleotide. For instance, a SNP in the initiator element nearly abolished the dominant max TSS of the protein coding gene *Smg9* in CAST (*Figure 4D*, *arrow*). Instead, initiation in CAST shifted to a new maxTSS located upstream (*Figure 4D*, *arrow head*), and also increased usage of several minor TSSs downstream (*Figure 4D*). This redistribution in both directions cannot be explained by the yeast unidirectional scanning model. Instead, we propose that after DNA melting and Pol II assembly on the template strand, Pol II rapidly and stochastically moves in both directions along the template strand scanning for an energetically favorable initiator dinucleotide in a process resembling Brownian motion (*Figure 4E*).

We also considered an alternative interpretation, that DNA sequence preference in the initiator dinucleotide feeds back and affects the stability of the PIC. Under this model, we expect that DNA sequence changes in a TSS would result in the redistribution of initiation signal to all of the potential TSSs within a TSR (assuming no change in the local concentration of components of Pol II, the preinitiation complex, or other core transcription factors). To examine this model, we asked whether the redistribution in initiation signal observed near mutant TSSs was also found in other TSCs within the TSR. We found that the increased initiation signal was limited to within ~20 bp of the CA/ non-CA initiation site (*Figure 4F*). In our view, this result supports the model proposed above, in which Pol II recruited by a single PIC initiates at an energetically favorable initiator dinucleotide within a confined region up to 20 bp in size.

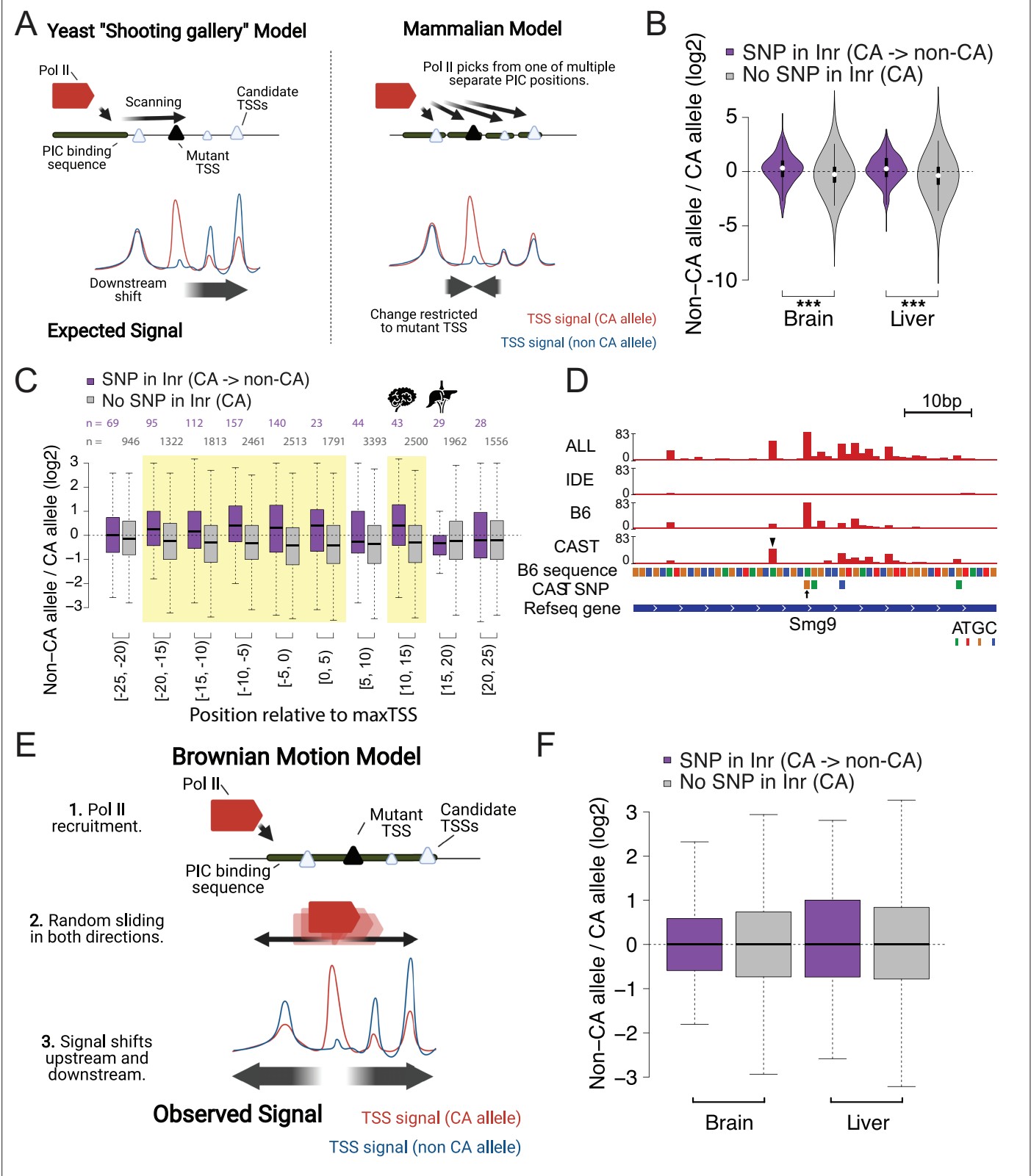

**Figure 4.** Brownian motion model of transcription start site selection. (**A**) Cartoon shows our expectation of the effects of allelic DNA sequence variation on transcription start site selection based on the yeast 'shooting gallery' model (left), or the mammalian model (right), in which Pol II initiates at potential TSSs (triangles) after DNA melting. We expected that mutations in a strong initiator dinucleotide (CA) on (for example) the CAST allele (bottom) would shift initiation to the initiator elements further downstream (yeast model), or would not affect the adjacent initiation sites (mammalian model). The size

*Figure 4 continued on next page*

*Figure 4 continued*

of the triangle indicates the strength of the initiator. (**B**) The violin plots show the distribution of ChRO-seq signal ratios on the candidate initiator motifs (including CA, CG, TA, TG) within 20 bp of the maxTSSs that had a CA dinucleotide in the allele with high maxTSS (SNP in Inr, purple) or had a CA dinucleotide in both alleles (No SNP in Inr, gray). Note that the central maxTSS was not included in the analysis. Wilcoxon rank sum test with continuity correction is p-value = 5.665e-10 for Brain and p-value <2.2e-16 for liver. (**C**) The box plots show the distribution of ChRO-seq signals ratios at TSSs with any YR dinucleotide (i.e. CA, CG, TA, TG) in both alleles as a function of the distance from the maxTSSs that had a CA dinucleotide in the allele with high maxTSS (SNP in Inr, purple) or had a CA dinucleotide in both alleles (No SNP in Inr, gray). Yellow shade indicates Wilcoxon Rank Sum and Signed Rank Tests (SNP in Inr vs no SNP in Inr) with fdr ≤ 0.05. The TSCs were combined from the brain and liver samples. (**D**) The browser shot shows an example of a maxTSS with increased initiation upstream and downstream of an allelic change in a CA dinucleotide. The arrow denotes a SNP at the maxTSS, in which B6 contains the high maxTSS with CA and CAST contains CG. The ChRO-seq signals at the alternative TSS with a CA dinucleotide (arrow head) upstream of the maxTSS were higher in the low allele (CAST in this case), resulting in a different maxTSS in CAST. Additional tracks show the B6 reference genome sequence, the position of all SNPs between B6 and CAST, and RefSeq gene annotations. All tracks line up with ChRO-seq data. ALL indicates signal from all reads, IDE indicates signal from reads that are not tagged with a SNP, B6 indicates signal from reads tagged with B6 SNP, CAST indicates signal from reads tagged with CAST SNP. (**E**) Proposed model in which Pol II initiates from a PIC and selects an energetically favorable TSS by random movement along the DNA similar to Brownian motion. (**F**) Boxplots show the distribution of ChRO-seq signal ratios on the candidate initiator motifs (including CA, CG, TA, TG) in the same TSR in the allele with high maxTSS (SNP in Inr, purple) or had a CA dinucleotide in both alleles (No SNP in Inr, gray). Note that the central maxTSS was not included in the analysis.

The online version of this article includes the following figure supplement(s) for figure 4:

**Figure supplement 1.** Violin plots show the distribution of ChRO-seq signals ratios between high low alleles at the candidate initiator motifs (All Inr: CA, CG, TA, TG; OnlyWeak Inr: CG, TA, TG; and OnlyCA) that are within 20 bp of the maxTSSs that had a CA dinucleotide in the allele with high maxTSS (SNP in Inr, purple) or had a CA dinucleotide in both alleles (No SNP in Inr, gray) in Brain(BN) or Liver(LV).

## Correspondence and disconnect between allele-specific TSS and pause position

We next examined allelic changes in the position of paused Pol II. To measure the position of the Pol II active site with single nucleotide precision, we prepared new ChRO-seq libraries in three organs (heart, skeletal muscle, and kidney from two female mice). New libraries were paired-end sequenced to identify the transcription start site and active site of the same molecule (*Tome et al., 2018*). New libraries clustered with those generated previously from the same organ (*Figure 5—figure supplement 1*). Using the same pipeline we developed for transcription initiation, we identified regions enriched for transcription initiation and pausing, and validated that candidate maxTSSs were enriched for the initiator motif (*Figure 5—figure supplement 2A*). Our analysis identified 2260 dREG sites with candidate changes in the shape of paused Pol II, assessed using the position of the Pol II active site, defined as the 3′ end of RNA insert (FDR ≤ 0.1; Kolmogorov-Smirnov [KS] test; see Materials and methods).

Previous work has shown a tight correspondence between the site of transcription initiation and pausing, with pausing occurring predominantly in the window 20–60 bp downstream of the TSS (*Tome et al., 2018*). As expected, allelic changes in the position of paused Pol II were often coincident with changes in transcription initiation (*Figure 5A*), particularly when the changes were large.

In addition to the main component of correlation between initiation and pausing, however, we also identified changes in both pause and initiation that were independent of the other step in the transcription cycle. In at least 111 cases (~31% of 359 total sites tagged by a SNP), the position of the max pause was identical between alleles, but the position of the max TSS changed by 1–32 bp (*Figure 5B*, *top*). Conversely, we identified 575 cases (~69% of 823 total sites tagged by a SNP) in which the Pol II pause position changed while both alleles shared the same max TSS (*Figure 5B*, *bottom*). Although changes in pause position occurred frequently, changes were mostly small in magnitude, typically separating the position at which paused Pol II was highest by less than 10 bp between alleles (*Figure 5B*, *bottom*). These cases in which the Pol II pause site changed without an accompanying change in the position of transcription initiation suggests the existence of a DNA sequence code that influences Pol II pausing.

## DNA sequence determinants of promoter proximal pause position

To understand the genetic determinants of pausing, we analyzed cases in which the same TSS had different maximal pause sites in the CAST and B6 alleles. As tagged SNPs in the window between the initiation and pause site were relatively rare, we increased our statistical power by analyzing all three

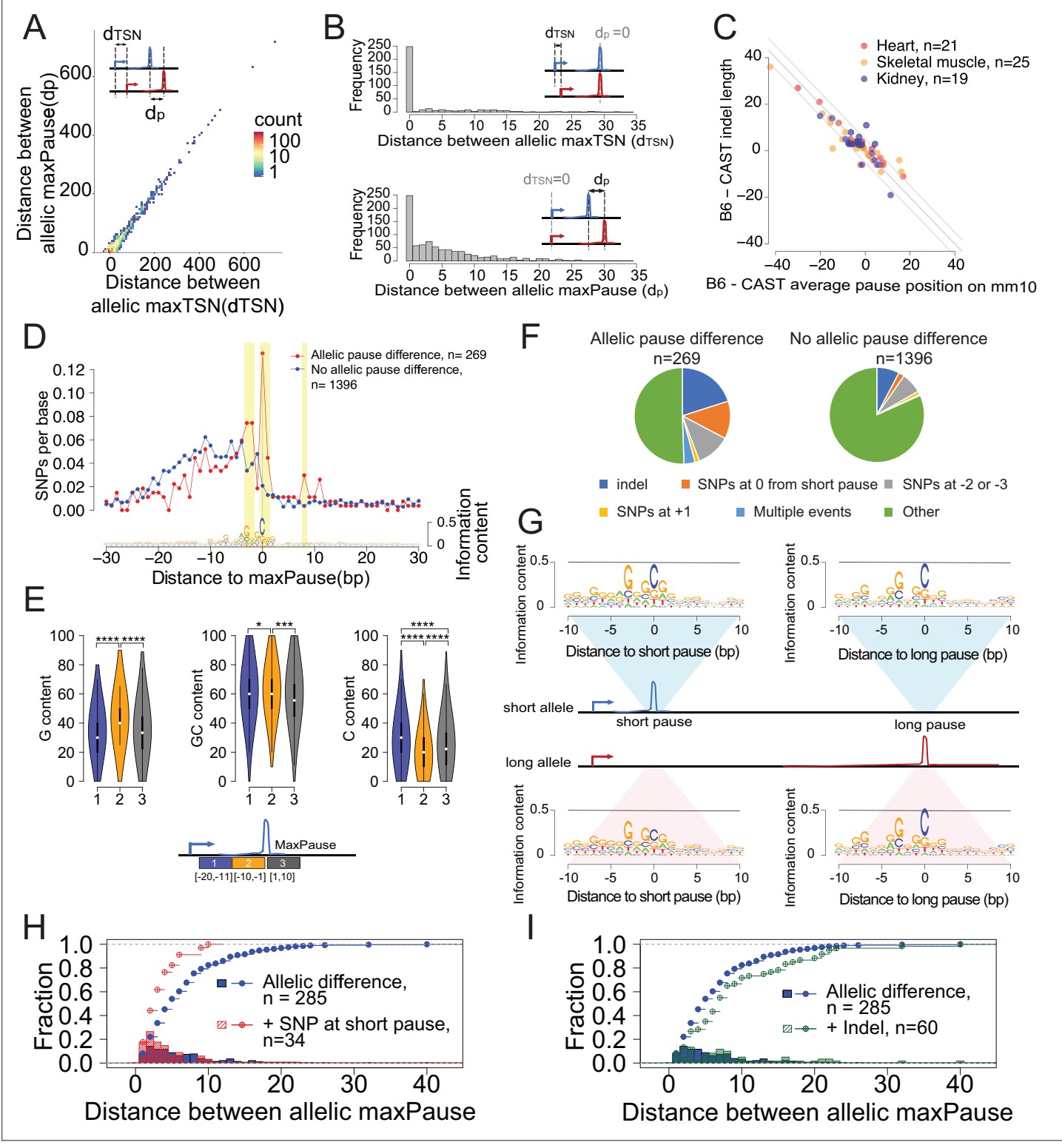

**Figure 5.** Allele-specific effects on the distribution of Pol II in the promoter proximal pause. (**A**) Scatterplots show the relationship between distances of allelic maxPause and allelic maxTSS within dREG sites with allelic different pause (n=2,260). (**B**) Top histogram shows the number of sites as a function of the distance between allelic maxTSS in which the allelic maxPause was identical (n=359). Bottom histogram shows the number of sites as a function of the distance between allelic maxPause where the allelic maxTSS was identical (n=823). (**C**) Scatterplot shows the relationship between indel length and the allelic difference of the average pause position on the reference genome (mm10). The pause positions of CAST were first determined in the CAST genome and then liftovered to mm10. Only sites initiated from the maxTSS and with allelic difference in pause shape were shown (KS test, fdr ≤ 0.1; also

*Figure 5 continued on next page*

*Figure 5 continued*

requiring a distinct allelic maximal pause). Color indicates the organs from which the TSS-pause relationship was obtained. (**D**) Top: scatterplot shows the average SNPs per base around the position of the Pol II in which the distance between the maxTSS and the max pause was lowest (short pause). Red represents sites with allelic difference in pause shape (Allelic pause difference, KS test, fdr ≤ 0.1 with distinct allelic maxPause, n=269), Blue is the control group (No allelic pause difference, KS test, fdr >0.9 and the allelic maxPause were identical, n=1,396). Bottom: The sequence logo obtained from the maxPause position based on all reads (n=3456 max pause sites). Sites were combined from three organs, after removing pause sites that were identical between organs. (**E**) Violin plots show the G content, GC content and C content as a function of position relative to maxPause defined using all reads (combined from three organs with duplicate pause sites removed, n=3456), block 1 was 11–20 bp upstream of maxPause, block 2 was 1–10 bp upstream of maxPause, and block 3 was 1–10 bp downstream of maxPause (* p<0.05, ** p<0.01, *** p<0.001, **** p<0.0001). Block 2 had a higher G content and a lower C content than the two surrounding blocks. (**F**) Pie charts show the proportion of different events around the maxPause of short alleles (short pause) with or without allelic pause differences. Sites were combined from three organs with duplicated pause sites removed. (**G**) Sequence logos show the sequence content of short alleles and long alleles at 270 short pause sites and 278 long pause sites. (**H**) The histograms show the fraction of pause sites as a function of distance between allelic maxPause, that is the distance between the short and long pause. The lines show the cumulative density function. Blue represents pause sites with allelic differences (n=285); red is a subgroup of blue sites with a C to A/T/G SNP at the maxpause (n=34). Two-sample Kolmogorov-Smirnov test p-value = 0.002694 (**I**) The histograms show the fraction of pause sites as a function of distance between allelic maxPause. The lines show the cumulative density function. Blue is pause sites with allelic differences (n=285), green is a subgroup of blue sites that contain indels between initiation and long pause sites (n=60), Two-sample Kolmogorov-Smirnov test p-value = 0.02755.

The online version of this article includes the following figure supplement(s) for figure 5:

**Figure supplement 1.** Spearman's rank correlation of the ChRO-seq data, including samples with a single base resolution for the Pol II active site.

**Figure supplement 2.** Validation of TSS identification in single-base run-on ChRO-seq libraries.

of the organs together after removing duplicate initiation sites when they overlapped. After filtering, we identified 269 candidate positions in which the same TSS gave rise to separate pause distributions on the B6 and CAST alleles. As a control, we used 1396 TSS/ pause pairs that were tagged by SNPs but did not have allelic changes in the pause position.

We first examined how short insertions or deletions affect the position of paused Pol II. Paused Pol II is positioned in part through physical constraints with TFIID, a core component of the PIC (*Fant et al., 2020*). As a result of such connections with the PIC, short insertions or deletions affecting the distance between the TSS and the maximal pause site altered the frequency of pausing at a model gene (*D. melanogaster HSP70*; *Kwak et al., 2013*). In our dataset, changes in pausing were highly enriched for small insertions and deletions between the maxTSS and pause site (n=56 (21%); expected = 22 (8%); p<1e-5, Fisher's exact test). Changes in pause position were correlated with changes in the size of the insertion or deletion, such that the number of bases between the max TSS and the pause was typically less than 5 bp (*Figure 5C*). Although this result may be influenced by fragment length bias introduced during sequencing, it nevertheless provides additional support for a model in which paused Pol II is placed in part through physical constraints with the PIC (*Fant et al., 2020*; *Kwak et al., 2013*).

Next we identified single-nucleotide changes that affect the position of the pause site. Previous studies have defined a C nucleotide at the paused Pol II active site (*Gressel et al., 2017*; *Tome et al., 2018*; *Watts et al., 2019*), which we recovered by generating sequence logos of max pause positions in our three murine organs (*Figure 5D*, *bottom*). Additionally, we also observed a G in the +1 position immediately after the pause, and a G/A-rich stretch in the 10 bp upstream of the pause site that lies within the transcription bubble (*Figure 5D*, *bottom*). The 10 bp upstream of the pause position had a higher G content and a lower C content than the two surrounding windows (*Figure 5E*). Thus, our data show that Pol II pauses on the C position immediately after a G-rich stretch.

Allelic differences at the RNA polymerase active site (usually a C) had the strongest association with the pause, followed by SNPs at the −2 and −3 position relative to the pause (usually a G; *Figure 5D*; *Figure 5—figure supplement 2B*). We also noted enrichment of SNPs downstream of the pause, especially in the +1 position, although the number of SNPs supporting these positions were small. We also noted that multiple independent SNPs were frequently found in the same TSS/ pause pair (observed n=10 (3.7%); expected = 1 (0.36%); <2e-5; Fisher's Exact Test; *Figure 5F*), suggesting that multiple changes in the weak DNA sequence motifs associated with pausing were more likely to affect the position of paused Pol II. Collectively, indels and SNPs identified as enriched in the analysis above explained 49% of allele-specific differences in the pause position (*Figure 5F*). Thus, the DNA sequence determinants of pause position are largely found either within the pause site, the transcription bubble of paused Pol II, or insertions or deletions between the pause and initiation site.

## Pol II pause position is driven by the first energetically favorable pause site

We extended our analysis of allelic differences in pausing to determine how multiple candidate pause positions early in a transcription unit collectively influence the position of paused Pol II. As in the analysis above, we focused on the set of allelic differences in pause shape in which the two alleles had a distinct maximal pause position (n=269). By definition, these transcription units had different maximal pause positions on the two alleles: on one allele the distance between the TSS and the pause position is shorter (which we call the 'short allele'), and on the other the distance between the TSS and the pause is longer ('long allele') (see cartoon in *Figure 5G*, *middle*). We found that the DNA sequence motif near the long pause position was similar on both alleles, recapitulating the C at the pause site and an enrichment of Gs in the transcription bubble (*Figure 5G*, *right*). By contrast, DNA sequence changes affecting pause position occurred at the short pause position on the long allele (*Figure 5G*; *bottom left*).

Our findings suggest a model in which single nucleotide changes that alter the free energy of the pause complex result in Pol II slipping to the next available position downstream. In favor of this model, when there was a SNP in the active site at the short pause, the max pause position moved downstream by <10 bp, a relatively small amount compared to all changes in pause shape (*Figure 5H*). By contrast, indels between the initiation and short pause site tended to have a larger effect on the allelic difference between pause positions (*Figure 5I*). These observations support a model in which DNA sequence changes that disfavor pausing result in Pol II slipping downstream to the next pause site for which DNA sequence is energetically favorable.

## Allelic changes in gene length caused by genetic differences in Pol II termination

Blocks with allelic bias identified by AlleleHMM were frequently found near the 3' end of genes. We hypothesized that these blocks reflected allelic differences in the position of Pol II termination that altered the length of primary transcription units. For example, *Fam207a* had an excess of reads on the CAST allele without a new initiation site that could explain allelic differences (*Figure 6A*). We set out to identify protein-coding transcription units that have an allelic difference in the abundance of Pol II only at the 3' (and not the 5') end.

To approximate the boundaries of allelic differences in Pol II abundance at the 3' end of genes, henceforth called the allelic termination window, we used AlleleHMM blocks that begin inside of a transcription unit and end near or after the end of the same transcription unit. We identified 317–931 candidate allelic termination windows in each of the eight organs (total n=3,450). Allelic termination windows varied in size between 1 kb and 100 kb, with a median size just under 10 kb (*Figure 6B*; *Figure 6—figure supplement 1A*). Although the longest allelic termination windows likely reflect allelic changes in multiple genes, the median size is approximately consistent with the reported length of transcription past the polyadenylation cleavage site (*Grosso et al., 2012*). Several lines of evidence suggest that these allelic differences were enriched for bona-fide differences in the site of Pol II termination: First, the majority did not start near dREG sites (the start is marked by a triangle in *Figure 6A*), and second, the allele with higher expression tended to have a similar ChRO-seq signal as the primary gene (*Figure 6—figure supplement 1B*).

Allelic differences in termination were consistent between different organs. For example, *Fam207a* had approximately the same allelic termination window in brain and liver (*Figure 6A*). To visualize the boundaries of allelic termination windows across larger numbers of transcripts, we used heat maps centered at the start of the allelic increase in expression (marked by a triangle) and sorted by the length of the allelic termination window (*Figure 6C*). Heatmaps showed a higher abundance of ChRO-seq reads in the allelic termination window on the allele with higher expression in the transcript body, as expected. Heat maps from brain and spleen using the same order as liver recovered similar patterns of allelic termination (*Figure 6C*). We conclude that allelic differences in termination were driven predominantly by a similar DNA sequence code across the set of organs analyzed in the present study.

Next, we examined how allelic termination windows were associated with changes in nearby transcription. Allelic termination windows were more likely to occur in highly expressed transcripts, which may partially reflect increased statistical power for detecting changes supported by larger numbers

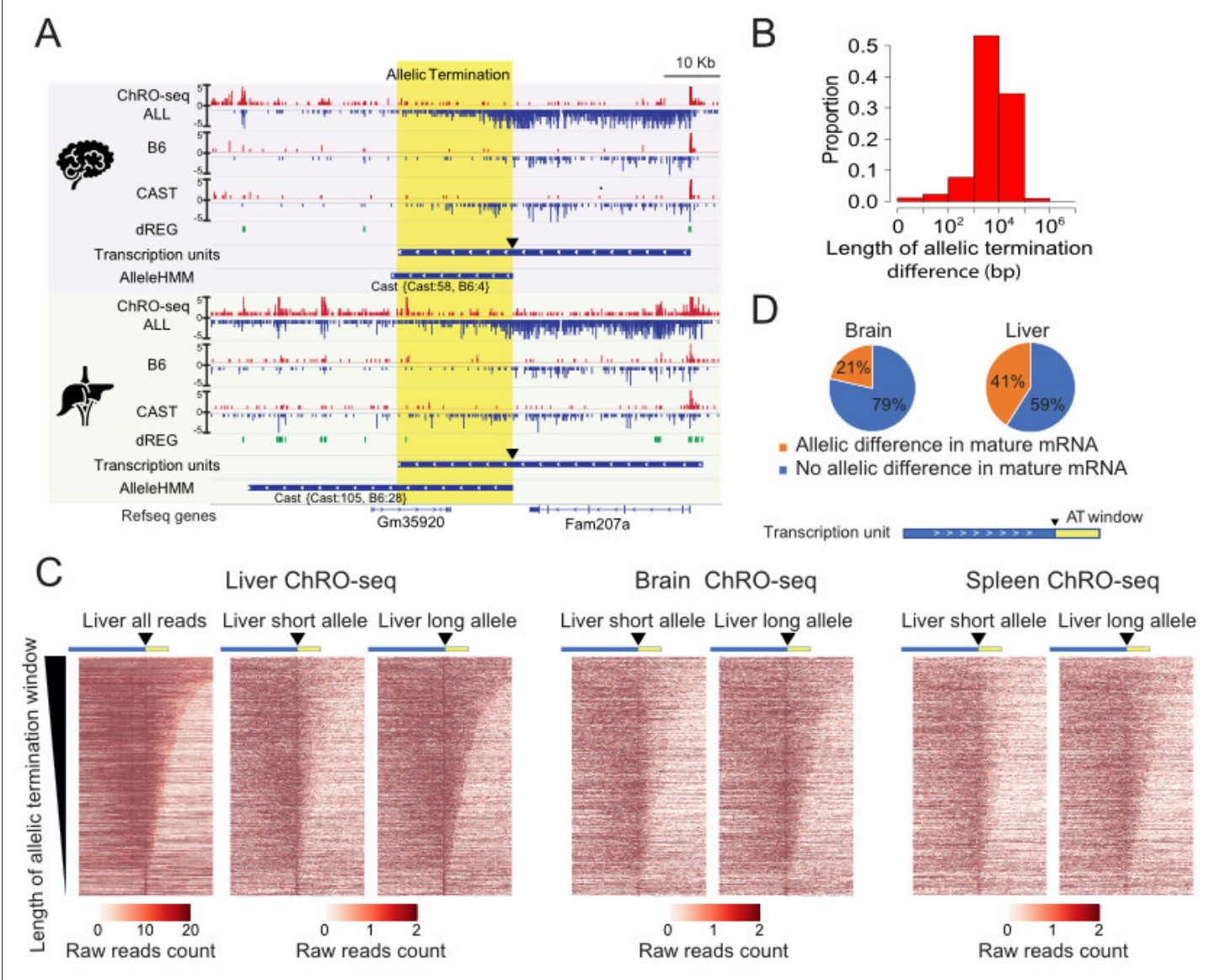

**Figure 6.** Widespread allele specific differences in the Pol II termination site. (**A**) The browser shot shows an example of allelic termination differences (yellow shade) in both brain and liver. Pol II terminates earlier on the B6 allele, resulting in a longer transcription unit on the CAST allele. The difference in allelic read abundance was identified by AlleleHMM. We defined the allelic termination difference (yellow shade) using the intersection between the transcription unit and AlleleHMM blocks. Tracks represent all ChRO-seq signal (top, marked ALL), reads mapping uniquely to the B6 or CAST allele (mid), the location of dREG, transcription units and AlleleHMM blocks (bottom). The position of the start of allelic termination is marked by an arrow. The allelic termination window is marked by yellow shading. (**B**) The histogram shows the fraction of transcription units as a function of the length of allelic termination difference. (**C**) Heatmaps show the raw read counts in transcription units (blue bar) with an allelic termination difference (yellow bar), centered at the beginning of allelic termination (solid triangle). The heatmap bin size is 500 bp, and 20 kb is shown upstream and downstream. The rows were sorted by the length of allelic termination differences determined by ChROseq signals from Liver. The short and long alleles were determined based on analysis of the liver. (**D**) Pie charts show the proportion of transcription units with allelic termination difference that also contains allelic difference in mature mRNA (orange).

The online version of this article includes the following figure supplement(s) for figure 6:

**Figure supplement 1.** Validation of allelic changes in termination.

**Figure supplement 2.** Allelic changes in 3′ exons correlated with mRNA stability.

of reads. Additionally, allelic termination windows were more likely to have allelic changes in the transcription level of an adjacent transcript compared with matched transcripts without an allelic termination window (liver: odds ratio = 1.65, p=1.83e-15; brain: odds ratio = 1.23, p=0.018; Fisher's exact test). We tested whether the association between allelic termination and adjacent transcript expression was also found when allelic termination and the adjacent transcript were encoded on opposite strands, hence avoiding the interpretation that AlleleHMM was more likely to detect allelic termination windows near an allelic difference with a large magnitude. The enrichments we observed were still significant in this more restrictive test (liver: odds ratio = 1.33, p=1.32e-05; brain: odds ratio = 1.16, p=0.09; Fisher's exact test). Intriguingly, the expression of nearby transcription units was frequently highest on the allele that terminated early (liver: odds ratio = 4.97, p=2.2e-16; brain: odds ratio = 9.67, p=2.2e-16; Fisher's exact test). Taken together, these results suggest an association between the length of post-poly-A transcription and the expression of nearby transcripts.

## Allelic changes in termination correlate with differences in mRNA primary structure and stability

To determine whether allelic differences in termination influence the primary structure (i.e. the processed mRNA sequence) of the mature mRNA, we next sequenced poly-A enriched mRNA from two liver and brain samples. Full-length mRNA-seq data can detect allelic differences in the frequency of use between exons within a gene based on the relative mRNA-seq signal in each exon. Differences occurring in exons at the 3' end of the gene are likely to reflect alternative use of polyadenylation cleavage sites between transcripts, a mechanism which has recently been reported to be subjected to genetic changes in humans (*Mittleman et al., 2021*; *Mittleman et al., 2020*). For instance, we identified an unannotated 3' exon in *Sh3rf3* that most likely reflects the use of a novel polyadenylation cleavage site unique to the allele having a longer transcription unit (*Figure 6—figure supplement 2A*). In most cases, including *Sh3rf3*, RNA-seq signal in the novel candidate exon was lower than in the annotated 3' exon, indicating that novel exons were included in only a portion of the mRNAs produced from the allele with the longer transcription unit.

To map allelic changes to mRNA primary structure involving unannotated candidate exons, we used AlleleHMM to identify differential mRNA abundance between alleles using the mRNA-seq data. Our analysis determined that 21–41% of transcription units with differences in allelic termination had evidence of changes in mRNA primary structure identified using AlleleHMM (*Figure 6D*), an enrichment of two- to fourfold relative to transcription units with no evidence of allelic termination (p<*2.2e-16*; Fisher's exact test in both brain and liver). We conclude that changes in allelic termination were often associated with corresponding changes in the primary structure of the mature mRNA.

Next, we examined whether changes in the mRNA primary structure affected the stability of the mRNA. We estimated mRNA stability in liver and brain using the ratio of signal in mRNA-seq and ChRO-seq data, which is correlated with the mRNA degradation rate (*Blumberg et al., 2021*). We asked whether transcription units with differences in both allelic termination and RNA primary structure had larger differences in mRNA stability between alleles compared with transcripts with no evidence of a difference in RNA primary structure. Allelic changes in mRNA primary structure were more likely to have increased differences in mRNA stability between B6 and CAST alleles (Brain p=4.28e-4, liver p=2.69e-08; KS test; *Figure 6—figure supplement 2B*-**C**). We emphasize that novel isoforms are likely to be a minor portion of the mRNA produced from the allele with the longer transcription unit, which could help explain why differences in mRNA stability occur at a relatively small portion of mRNAs with changes in primary structure. Thus, differences in allelic termination were frequently associated with changes in the primary structure of the mature mRNA and in some cases impact mRNA stability.

## Discussion

Despite much progress, we still have a relatively limited understanding of how DNA sequence influences the process of transcription initiation, pause, elongation, and termination by RNA Pol II. Our limited understanding reflects, in part, the fact that DNA sequence elements that influence transcription are highly degenerate and frequently spread across wide genomic regions, making them very difficult to identify and characterize. Here, we used naturally occurring genetic variation in F1

hybrids of two highly divergent mouse strains, CAST and B6. We generated and analyzed ChRO-seq data, which maps the location and orientation of RNA polymerase, in eight murine organs. This data provides a window into how short SNPs and indels influence the position of Pol II that is robust to technical variation, providing new insights into Pol II dynamics.

We characterized the effects of DNA sequence variation on the position of the TSS during Pol II initiation. DNA sequence differences within ~5–10 bp of the TSS were the most common determinant of the TSS. As expected, the initiator element plays a central role. The A base in the initiator motif, in particular, was the most important determinant of the TSS. Changes in the cytosine nucleotide at the −1 position had surprisingly little evidence for differences in our analysis, consistent with a hierarchy of transcription initiation that is more dependent on the A in a CA initiator dinucleotide. Unexpectedly, we found that AT nucleotides surrounding the initiation site were positively correlated with the use of that TSS between alleles. We speculate that AT content influences the TSS because AT-rich DNA requires lower free energy to melt (*Breslauer et al., 1986*). Thus, our study reveals both known and novel DNA sequence elements surrounding the TSS that influence which of the numerous potential TSSs within the TSC are selected for initiation.

Our results have led us to propose a new model of transcription initiation in which Pol II selects an initiation site through random motion on DNA resembling Brownian motion. In the prevailing model of initiation in yeast, called the 'shooting gallery model', DNA is melted and Pol II scans by forward translocation until it identifies an energetically favorable TSS (*Braberg et al., 2013*; *Kaplan et al., 2012*; *Qiu et al., 2020*). Mammals are not believed to scan, but rather independent PICs are believed to support initiation from a narrow window (*Luse et al., 2020*). Although our study does support a role for DNA sequence motifs which control the location of the PIC in selecting the TSS, PIC motifs such as the TATA box were relatively less important for specifying the exact position of the TSS. Intriguingly, DNA sequence changes in the initiator element increased the use of nearby TSSs both upstream and downstream in a window of ~20 bp. We think the most straightforward interpretation of these results is that Pol II samples candidate initiation sites in both directions by a process resembling a one-dimensional Brownian motion along the DNA template.

Another possible interpretation is that changes in the CA dinucleotide sequence alter DNA elements that support partially overlapping PICs (e.g. CA ->TA dinucleotide change makes the sequence slightly closer to a TATA box that could support initiation further downstream). Although we do not directly rule out this alternative interpretation, in our view it seems less likely because the effects we observe are constrained within a fairly narrow window, because the effects we observed are so consistent when conditioning on changes in the initiator DNA sequence motif, and because DNA sequence changes in the TATA box did not have as large of an impact on the initiation site. In either case, however, it is clear from our data that changes in the DNA sequence of initiator elements tend to increase the use of candidate initiators nearby.

Our data also suggest that the region in which Brownian motion selects a TSS may be as large as 20 bp. It seems likely that the ~20 bp distance limitation we observed reflects structural constraints placed on Brownian motion by the PIC or the RNA Polymerase complex. We caution, however, that our analysis does not account for the location of PIC binding relative to the location of the mutant initiator. Cases where the mutant initiator happens to be located near the edge of the structural constraint will allow use of new TSSs further from the mutant initiator, potentially increasing the width of the window we observed in our ensemble analysis. New studies integrating precise information about the location of PIC binding, using ChIP-exo/ nexus or equivalent assays, with allele-specific TSSs will be necessary to unravel how the PIC or other protein components constrain the location of Brownian motion in TSS selection.

By analyzing allelic changes in pause position from the same initiation site, we have learned much about how DNA sequence influences the precise coordinates of promoter proximal pause. Our results show that the pause position occurs at a C nucleotide downstream of a G-rich stretch, similar to motifs enriched at pause positions in prior studies (*Gressel et al., 2017*; *Tome et al., 2018*). As cytosine is the least abundant ribonucleotide (*Traut, 1994*), previous authors proposed it is the slowest to incorporate into nascent RNA, explaining its association with the Pol II pause (*Tome et al., 2018*). In addition, we also identified a G-rich stretch that coincides with the position of the transcription bubble, as well as a guanine nucleotide just downstream of the pause position. The enrichment of G nucleotides in the transcription bubble has a higher stability of RNA-DNA hybridization without using a cytosine

nucleotide, and may serve to stabilize the RNA-DNA hybrid within the transcription bubble while Pol II remains paused.

We also noted widespread allelic differences in the site of transcription termination, which resulted in substantial differences in the length of primary transcription units between alleles. Allelic differences in transcription termination were largely similar between different organs, implicating a single DNA sequence code as the major determinant of transcription termination across organs studied here. Although the majority (60–80%) of allelic differences in termination did not affect the primary structure (i.e. the set of all ribonucleotides in the mRNA), they often did have an impact that we were able to detect using full-length mRNA-seq data. We note that in cases where the inclusion of a novel exon appeared to be altered between two alleles, the novel isoform typically represented only a small portion of the total mRNA population, as judged by comparing the signal in the alternative and annotated 3' exons. Nevertheless, differences in mRNA stability between alleles were more common in cases where alternative termination altered mRNA primary structure. These findings suggest that changes in Pol II termination can impact mRNA primary structure and ultimately mRNA stability.

Our study used an F1 hybrid cross between CAST and B6, which we contend is particularly well suited to problems in which the DNA sequence determinants are weak and spread across a large genomic region. Experimental batch variation is a major confounder in all genomic analyses. However, batch effects are shared between the B6 and CAST alleles, which are processed from the same nucleus and undergo exactly the same experimental processing and sequencing steps. Another advantage of using a F1 hybrid system is that by comparing the effect of SNPs or indels on otherwise very similar DNA sequences, we can reduce artifacts from larger changes in DNA sequence composition on the sequencing library itself. Intuitively, DNA sequence changes affecting the middle of an RNA insert, especially those which constitute only a single base difference between alleles, are unlikely to have a large impact on detection in ChRO-seq. As a result, we can be confident in differences observed in our analysis, even in cases where the differences between the CAST and B6 alleles were relatively small in magnitude.

The use of an F1 hybrid system does, of course, have a number of limitations as well. Any DNA sequence variation between alleles that affects the probability of detection, including either DNA sequence variation or size bias in the RNA insert, could impact detection differently between alleles. One place where this might occur is ligation bias for SNPs on the 5' or 3' end of an RNA insert. However, we think the effect of such sequence bias is relatively small in our study. Results that might be affected, including the A base in the initiator element (at the 5' end of the RNA insert), and the C base in the Pol II active site (at the 3' end of the RNA insert), are supported by existing literature (*Kaufmann and Smale, 1994*; *Kuehner and Brow, 2006*; *Smale and Baltimore, 1989*; *Tome et al., 2018*). Major new results reported here generally reflect SNPs inside of the RNA insert, making differences in detection a less likely explanation.

In summary, our study dissects how DNA sequences impact steps during the Pol II transcription cycle. The dynamics of Pol II on each position of the genome can influence the rate of mRNA production and may impact organism phenotypes. Our work is a first step in understanding how DNA sequences influence stages in the Pol II transcriptional cycle and impact phenotypes in humans and other animals.

## Materials and methods
### Experimental methods
### Mouse experiments

The mice used in this study were reciprocal F1 hybrids of the strains C57BL/6 J and CAST/EiJ. All mice were bred at Cornell University from founders acquired from the Jackson Laboratory. All mice were housed under strictly controlled conditions of temperature and light:day cycles, with food and water ad libitum. All mouse studies were conducted with prior approval by the Cornell Institutional Animal Care and Use Committee, under protocol 2004–0063.

### Tissue collection

Mice were euthanized at 22–25 days of age by $CO_2$, followed by cervical dislocation. All mice were euthanized between 10 a.m. and 12 p.m., immediately after removal from their home cage. Whole

brain, eye, liver, stomach, large intestine, heart, skeletal muscle, kidney, and spleen were rapidly dissected and snap frozen in dry ice.

## mRNA isolation/ RNA-seq library prep

RNA was extracted from the brain and liver of a male and a female mouse (both 22d of age). Tissue samples were frozen using liquid nitrogen and pulverized using a mallet and a mortar. 100 mg of each tissue was used for a TRIzol RNA extraction. Briefly, 1 mL of TRizol was added to each sample, chloroform was used for phase separation of the aqueous phase containing RNA, RNA was precipitated using isopropanol and washed with 75% ethanol. A total of 400 ng of RNA was input into the RNA-seq library prep. Poly-A containing mRNA was enriched for 2 rounds using the NEBNext Poly(A) mRNA Magnetic Isolation Module. Stranded mRNA-seq libraries were prepared by the Cornell TREx facility using the NEBNext Ultra II Directional RNA Library Prep Kit. Libraries were sequenced using an Illumina NextSeq500.

## Chromatin isolation

Chromatin was isolated and ChRO-seq libraries were prepared following the methods introduced in our recent publication (*Chu et al., 2018*). Briefly, tissue was cryo-pulverized using a cell crusher (http://cellcrusher.com). Tissue fragments were resuspended in NUN buffer (0.3 M NaCl, 1 M Urea, 1% NP-40, 20 mM HEPES, pH 7.5, 7.5 mM MgCl2, 0.2 mM EDTA, 1 mM DTT, 20 units per ml SUPERase In Rnase Inhibitor(Life Technologies, AM2694), 1×Protease Inhibitor Cocktail (Roche, 11 873 580 001)). Samples were vortexed vigorously before the samples were centrifuged at 12,500 x g for 30 min at 4 °C. The NUN buffer was removed and the chromatin pellet washed with 1 mL 50 mM Tris-HCl, pH 7.5 supplemented with 40 units of RNase inhibitor. Samples were centrifuged at 10,000 x g for 5 min at 4 °C and the supernatant discarded. Chromatin pellets were resuspended in storage buffer (50 mM Tris-HCl, pH 8.0, 25% glycerol, 5 mM Mg(CH3COO)2, 0.1 mM EDTA, 5 mM DTT, 40 units per ml Rnase inhibitor) using a Bioruptor sonicator. The Bioruptor was used following instructions from the manufacturer, with the power set to high, a cycle time of 10 min (30 s on and 30 s off). The sonication was repeated up to three times if necessary to resuspend the chromatin pellet. Samples were stored at –80 °C.

## ChRO-seq library preparation

ChRO-seq libraries were prepared following a recent protocol (*Mahat et al., 2016b*). We prepared some libraries to achieve single-nucleotide resolution for the Pol II active site. In these cases, the chromatin pellet was incubated with 2 x run-on buffer (10 mM Tris-HCl, pH 8.0, 5 mM MgCl2,1 mM DTT, 300 μuM KCl, 20 uM Biotin-11-ATP (Perkin Elmer, NEL544001EA), 200 uM Biotin-11-CTP (Perkin Elmer, NEL542001EA), 20 μM Biotin-11-GTP (Perkin Elmer, NEL545001EA), 200 μM Biotin-11-UTP (Perkin Elmer, NEL543001EA)) for 5 min at 37 °C. In some libraries, we modified the run-on buffer to extend the length of reads for more accurate allelic mapping at the expense of single nucleotide resolution for the Pol II active site. In these cases, the run-on reaction was performed using a different ribonucleotide composition in the nuclear run-on buffer (10 mM Tris-HCl, pH 8.0, 5 mM MgCl2,1 mM DTT, 300 mM KCl, 200 μM ATP (New England Biolabs (NEB), N0450S), 200 μ M UTP, 0.4 μ M CTP, 20 μ M Biotin-11-CTP (Perkin Elmer, NEL542001EA), 200 μ M GTP (NEB, N0450S)).The run-on reaction was stopped by adding Trizol LS (Life Technologies, 10296–010) and RNA was pelleted with the addition of GlycoBlue (Ambion, AM9515) to visualize the RNA. RNA pellet was resuspended in diethylpyrocarbonate (DEPC)-treated water. RNA was heat denatured at 65 °C for 40 s to remove secondary structure. RNA was fragmented using base hydrolysis (0.2 N NaOH on ice for 4 min). RNA was purified using streptavidin beads (NEB, S1421S) and removed from beads using Trizol (Life Technologies, 15596–026). We ligated a 3' adapter ligation using T4 RNA Ligase 1 (NEB, M0204L). We performed a second bead binding followed by a 5' decapping with RppH (NEB, M0356S). RNA was phosphorylated on the 5' end using T4 polynucleotide kinase (NEB, M0201L) then ligated onto a 5' adapter. A third bead binding was then performed. The RNA was then reverse transcribed using Superscript III Reverse Transcriptase (Life Technologies, 18080–044) and amplified using Q5 High-Fidelity DNA Polymerase (NEB, M0491L) to generate the ChRO-seq libraries. Libraries were sequenced using an Illumina HiSeq by Novogene. All adapter sequences and barcodes used for each sample are depicted in *Supplementary file 4*.

## Data analysis

### Read mapping, transcription start site, and transcription unit discovery

#### Processing and mapping ChRO-seq reads

Paired-end reads with single nucleotide precision were processed and aligned to the reference genome (mm10) with the proseq2.0 (https://github.com/Danko-Lab/proseq2.0 swh:1:rev:c3260bdffb-571beb58c33ea086a968d7ac519e6f)(*Chou, 2022*). Libraries in which we tailored the run-on to extend the length of reads were pre-processed, demultiplexed, and aligned to the reference genome (mm10) with the proseqHT_multiple_adapters_sequencial.bsh. AlleleDB (*Chen et al., 2016*; *Rozowsky et al., 2011*) align the R1 reads to the individual B6 and Cast genomes. In brief, the adaptor sequences were trimmed with the cutadapt, then PCR duplicates were removed using unique molecular identifiers (UMIs) in the sequencing adapters with prinseq-lite.pl (*Schmieder and Edwards, 2011*). The processed reads were then aligned with BWA (mm10) in analyses not using individual genome sequences (*Li and Durbin, 2009*), or with bowtie (*Langmead et al., 2009*) as input for AlleleDB. When bowtie was used, we selected either the R1 or R2 files for alignment for analyses requiring either the 5' or 3' end of the RNA insert. All scripts for mapping can be found publicly at: https://github.com/Danko-Lab/F1_8Organs/blob/main/00_F1_Tissues_proseq_pipeline.bash, https://github.com/Danko-Lab/utils/blob/master/proseq_HT/proseqHT_multiple_adapters_sequencial.bsh; (*Danko, 2019a*).

#### Processing and mapping RNA-seq reads

We used STAR (*Dobin et al., 2013*) to align the RNA-seq reads. To avoid bias toward the B6 genome, we did not use any gene annotations for mapping, but used the list of splicing junctions generated by STAR. Mapping was performed in three stages: First, reads were first mapped without annotation and STAR generated a list of splicing junctions (sj1) from the data. Second, to identify potential allele specific splicing junctions, we performed allele specific mapping using STAR which takes as input a VCF file denoting SNPs differentiating Cast and B6, using the initial splice junction list (sj1). This personalized mapping was used to generate a more complete list of splice junctions (sj2). Third, we identified allele specific alignments by using the WASP option provided by STAR (*van de Geijn et al., 2015*). In this final mapping, we used the splice junction list (sj2) and a VCF file. This procedure generated a tagged SAM file (vW tag) providing the coordinates of allele specific alignments and their mapping position. Scripts can be found here: https://github.com/Danko-Lab/F1_8Organs/blob/main/termination/F1_RNAseq_forManuscript.sh *dREG:* For each organ, we merged all reads from each replicate and cross to increase the power of dREG. BigWig files representing mapping coordinates to the mm10 reference genome were uploaded to the dREG web server at http://dreg.dnasequence.org (*Wang et al., 2019*). All the output files were downloaded and used in subsequent data analysis. Scripts used to generate the BigWig files can be found at: https://github.com/Danko-Lab/F1_8Organs/blob/main/F1_TSN_Generate_BigWig.sh.

#### Transcript unit prediction using tunits

We used the tunit software to predict the boundaries of transcription units de novo (*Danko et al., 2018*). We used the 5 state hidden Markov model (HMM), representing background, initiation, pause, body, and after polyadenylation cleavage site decay from tunits. To improve sensitivity for transcription unit discovery in each tissue, the input to tunits was the output of dREG and bigWig files that were merged across all replicates and crosses. Scripts can be found here: https://github.com/Danko-Lab/F1_8Organs/blob/main/Tunit_predict_manuscript.sh https://github.com/Danko-Lab/F1_8Organs/blob/main/run.hmm.h5_F1bedgraph.R; https://github.com/Danko-Lab/F1_8Organs/blob/main/hmm.prototypes.R.

#### Clustering

We used all transcripts that are 10,000 bp long from GENCODE vM25. Only reads mapped to the gene body (500 bp downstream of the start of the annotation to the end of the annotation) were used. We filtered the transcripts and only kept those with at least 5 mapped reads in every sample. We exported rpkm normalized expression estimates of each transcript. Morpheus was used to calculate and plot Spearman's rank correlation (https://software.broadinstitute.org/morpheus) with the following parameters: Metric = Spearman rank correlation, Linkage method = Average Linkage,

distance.function.name=Spearman rank correlation. Scripts can be found here: https://github.com/Danko-Lab/F1_8Organs/blob/main/getCounts_skipfirst500.R.

Allele specific heatmaps in *Figure 1C* focused on all genes that are longer than 10,000 bp from GENCODE vM25 and were allele specific in at least 3 of the samples from each organ. Chromosomes M, X, and Y were excluded from the analysis. We computed the log-2 ratio of reads mapping uniquely to the B6 and CAST allele for each gene. Log-2 ratios were used as the input to Morpheus. Rows (representing genes) were ordered by 1 - Pearson's correlation across all samples. Columns were ordered manually. Organs used the same order as in the total gene expression clustering, above. Samples were ordered based on the direction of cross so that imprinted genes could easily be distinguished from strain-effect genes.

## Testing positional correlation between transcripts

We asked whether adjacent transcription units shared the same allele specificity more frequently than chance. We identified transcription units that were allele specific based on a false discovery rate corrected binomial test cutoff less than 0.1. As a control set, we used all transcription units regardless of allelic bias. For each transcription unit in the allele specific and control set, we identified the number with a FDR corrected binomial test for allele specificity <0.1 (representing allele specific) or >0.9 (representing confident non-allele specific). The differences between groups were tested using a Fisher's exact test.

## AlleleHMM

Maternal- and paternal- specific reads mapped using AlleleDB were used as input to AlleleHMM (*Chou and Danko, 2019*). We combined biological replicates from the same organ and cross, and used the allele-specific read counts as input to AlleleHMM. AlleleHMM blocks were compared with GENOCODE gene annotations to pick the free parameter, $\tau$, which maximized sensitivity and specificity for computing entire gene annotations, as described (*Chou and Danko, 2019*). Most organs used a $\tau$ of either 1E-5 (brain, liver, spleen, skeletal muscle) or 1E-4 (heart, large intestine, kidney, and stomach). As reported, the primary parameter that influenced $\tau$ was the library sequencing depth. AlleleHMM scripts can be found here: https://github.com/Danko-Lab/AlleleHMM, (*Danko, 2019b*); https://github.com/Danko-Lab/F1_8Organs/blob/main/01_F1Ts_AlleleHMM.bsh.

## Discovering strain effect and imprinted domains

We used the following rules to merge nearby allele specific transcription events into strain effect or imprinted domains:

1. Identify candidate AlleleHMM blocks using pooled ChRO-seq reads from samples with the same organ and same cross direction.
2. Combine blocks above from the same organ (but different crosses). Combine p-values using Fisher's method for all biological replicates within the same tissue and cross direction. Keep blocks that are biased in the same direction with a Fisher's p-value ≤ 0.05.
3. Determine whether the blocks are under a strain effect (allelic biased to the same strain in reciprocal crosses) or parent-of-origin imprinted effect (allelic biased to the same parent in reciprocal crosses).
4. Merge overlapping strain effect blocks from different organs into strain effect domains; merge overlapping strain effects from the same imprinted blocks into imprinted domains.

Scripts implementing these rules can be found here: https://github.com/Danko-Lab/F1_8Organs/blob/main/Find_consistent_blocks_v3.bsh.

After discovering blocks, we examined the number of gene annotations in each domain (*Figure 1E*). We used GENCODE annotated genes (vM25). We kept all gene annotations and merged those which overlapped or bookended (directly adjacent to, as defined by bedTools) on the same strand so that they were counted once. All operations were performed using bedTools (*Quinlan and Hall, 2010*). Scripts can be found here: https://github.com/Danko-Lab/F1_8Organs/blob/main/Find_consistent_blocks_v3.bsh; https://github.com/Danko-Lab/F1_8Organs/blob/main/Imprinted_figures.R.

## Determining the allelic bias state of annotated genes

We used GENCODE gene annotations representing protein-coding genes (vM20) in which the transcription start site overlapped a site identified using dREG (*Wang et al., 2019*). We used de novo

annotations by the *tunits* package to identify unannotated transcription units, which do not overlap an annotated, active gene as a source of candidate transcribed non-coding RNAs. Transcription units from both sources were merged for downstream analysis. We determine if the gene/ncRNA are allelic biased by comparing mapped reads to the B6 and CAST genomes using a binomial test, retaining transcription units with a 10% false discovery rate (FDR). Allele specific transcription units were classified as being under a strain effect (allelic biased to the same strain in reciprocal crosses) or parent-of-origin imprinted effect (allelic biased to the same parent in reciprocal crosses). Scripts can be found: https://github.com/Danko-Lab/F1_8Organs/blob/main/Genetics_or_imprinting_v2.bsh.

## Evaluate the contribution of false negatives to organ-specific allelic bias in organ-specific allelic biased domains (OSAB domain)

In *Figure 1—figure supplement 1A and B*, we asked whether organs in which we did not identify allelic bias were false negatives. To do this we compared distributions of the transcription level in putatively unbiased organs. For each OSAB domain identified in at least one, but not in all organs, we examined the effect size of allelic bias in the organ with the highest expression that is putatively unbiased. We defined the effect size of allelic bias as the ratio between maternal and paternal reads in the candidate OSAB domain. If the allelic-biased organ was maternally biased, the effect size was calculated as maternal reads divided by paternal reads in the blocks, otherwise the effect size was calculated as paternal reads divided by maternal reads in the blocks. Scripts implementing this can be found here: https://github.com/Danko-Lab/F1_8Organs/blob/main/AllelicBiase_expressionLevel. bsh; https://github.com/Danko-Lab/F1_8Organs/blob/main/getNonBiasedHighest_Biased_AllelicBiaseDistribution.R.

## Evaluate the contribution of expression to organ-specific allelic biased domains (OSAB domain)

In *Figure 1—figure supplement 1C*, we asked whether OSAB domains were not actively transcribed in candidate unbiased organs. Using bedtools and in-house scripts, we calculated the rpkm (Reads per kilobase per million mapped reads) normalized transcription level of each strain effect block located within the OSAB domains in each organ. The full diploid genome was used for mapping. The non-allelic-biased organs with highest rpkm (nonBiasedH) were selected to compare with the rpkm of the allelic-biased organs in OSAB domains. Scripts implementing this can be found here: https://github.com/Danko-Lab/F1_8Organs/blob/main/AllelicBiase_expressionLevel.bsh; https://github.com/Danko-Lab/F1_8Organs/blob/main/getNonBiasedHighest_Biased_TotalReadCountRatio.R.

## Analysis of allele-specific initiation

### Identification of candidate transcription initiation sites

We used 5 prime end of ChRO-seq reads (the R1 paired-end sequencing file, which represents the 5 prime end of the nascent RNA) to identify candidate initiation sites using methods adapted from *Tome et al., 2018*. Briefly, candidate transcription start sites (TSS) from each read were merged into candidate transcription start clusters, in which the max TSS was identified. We identified candidate TSSs that fall within dREG sites and were supported by at least 5 separate reads. TSSs within 60 bp of each other were merged into candidate TSCs. The TSS with the maximal read depth in each TSC was defined as the maxTSS for that TSC. We allow each TSC to have more than one maxTSSs if multiple TSSs share the same number of read counts in that TSC. To test whether the candidate maxTSSs represented bona-fide transcription start sites, we generated sequence logos centered on the maxTSS using the seqLogo R package (*Bembom and Ivanek, 2019*). We retained tissues in which the maxTSS contained a clearly defined initiator dinucleotide that reflects a similar sequence composition as those previously reported (*Tome et al., 2018*). Additionally, we used an in-house R script to examine the relationship between TSS counts and Read counts of the TSC (*Figure 2—figure supplement 1B*), and found a similar relationship to those reported (*Tome et al., 2018*).

### Identify allele-specific differences in TSCs abundance (ASTSC abundance)

We used a binomial test to identify candidate allele-specific transcription start clusters, with an expected allelic ratio of 0.5. We filtered candidate allele-specific differences using a false discovery rate (FDR) corrected p-value of 0.1, corresponding to an expected 10% FDR.

## Identify allele-specific differences in TSC shape (ASTSC shape)

We used a Kolmogorov-Smirnov (K-S) test to identify TSCs where the distribution of transcription initiation differed significantly between the B6 and CAST alleles (ASTSC shape). We used TSCs with at least 5 mapped reads specific to the B6 genome and at least 5 mapped reads specific to CAST. Only autosomes were used. We corrected for multiple hypothesis testing using the false discovery rate (**Storey and Tibshirani, 2003**) and filtered ASTSC shapes using a 10% FDR. Applying the same criteria and K-S test to biological replicates (mice F5 and F6), identified fewer candidate differences between biological replicates compared with differences between B6 and CAST alleles in the related task of testing pause shape (0.02–2.5% of the universe of sites between biological replicates, compared with 12–16% of the universe of sites observed between B6 and CAST alleles). We further separated the ASTSC shape candidates into two groups: one driven by a single TSS (single TSS driven ASTSC shape), the other reflecting changes in more than one base in the TSC (multiple TSS driven ASTSC shape). To separate into two groups, we masked the TSS with the highest allelic difference (determined by read counts) within each TSC and performed a second K-S test. Multiple TSS-driven ASTSC were defined as those which remained significantly different by K-S test after masking the position of highest allelic difference. Single TSS driven ASTSCs were defined as ASTSCs that were no longer significantly different by K-S test after masking the maximal position. In the second K-S test, we used the nominal p-value defined as the highest nominal p-value that achieved a 10% FDR during the first K-S test.

## SNP analysis

We examined the distribution of single-nucleotide polymorphisms (SNPs) near ASTSCs from each class. A major confounding factor in SNP distribution is the ascertainment bias of requiring at least one tagged SNP to define the allelic imbalance between the two alleles, resulting in an enrichment of SNPs within the read. To control for this bias, we compared the set of sites with a significant change in the TSC shape or abundance (FDR ≤ 0.1) with a background control set defined as candidate TSCs in which there was no evidence of change between alleles (FDR >0.9) in all analyses. We display a bin size of 5 bp. To test for differences, we merged adjacent bins by using a bin size of 10 bp to increase statistical power and tested for enrichment using Fisher's exact test, FDR cutoff = 0.05. (**Figure 3E and F**). We also examined the difference in base composition between the allele with high and low initiation in each ASTSC shape difference centered on the position of the maxTSS in the allele with high initiation (in **Figure 3G**). We determined the high/low allele based on the transcription level at maxTSS. If there are more than one TSSs with the max read counts, there will be more than one maxTSSs representing each TSC.

## Comparison of SNPs in TATA motifs

We extracted the DNA sequence in the region between –35 and –20 bp upstream of each max TSN on both the B6 and CAST alleles. We used RTFBSDB (**Wang et al., 2016**) to compute the maximal score (defined using the log likelihood ratio of a motif match compared to a 1 bp Markov model as background) of a TATA motif (>3) in this region using two TATA motifs: M00216 (low information content) and M09433 (high information content). We compared the difference in the maximal score between the B6 and CAST alleles to the difference in max TSN usage between B6 and CAST alleles. Pearson's correlation between these two variables was tested using the cor.test function in R.

## Comparison of AT content between alleles

As a proxy for melting temperature (in **Figure 3H**), we examined the AT content in 5 bp windows around the maxTSS on alleles with high and low maxTSS usage. As in the SNP analysis (above), we compared the set of sites with a significant change in the TSC shape or abundance (FDR ≤ 0.1) with a background control set defined as candidate TSCs in which there was no evidence of change between alleles (FDR >0.9). Computations were performed using R library TmCalculator (**Li, 2019**). We used Fisher's exact test to examine if there was an enrichment of AT(in the high allele) to GC (in the low allele) SNPs in each 5 bp bin, and adopted an FDR corrected p-value cutoff = 0.05. In all analyses, positions at –1 and 0 relative to the maxTSS were masked to avoid confounding effects of the initiator sequence motif on computed AT content.

## Shooting gallery

In our analysis of the shooting gallery model, we focused on a subset of TSCs which do not appear to change expression globally, and which have an SNP in the initiator element. Toward this end, we identified TSCs which do not overlap AlleleHMM blocks. We set the allele with high and low expression based on allele-specific reads in the maxTSS. Next, we divided data into a test and background control dataset in which the test set had a CA dinucleotide in the allele with high maxTSS use and any other combination except for CA on the other allele. The control set did not have a CA dinucleotide in the maxTSS initiator position. Next we computed the distance to the maxTSS and the allelic read count at other candidate initiator motifs (including CA, CG, TA, TG). In all analyses, we compared the set of maxTSSs with SNPs in the initiator position with the control set which did not have a SNP. Statistical tests used an unpaired Wilcoxon rank sum test. We corrected for multiple hypothesis testing using false discovery rate.

All scripts implementing analysis of allele-specific initiation can be found at: https://github.com/Danko-Lab/F1_8Organs/tree/main/initiation.

## Analysis of allele-specific pause

### Identification of allele-specific differences in pause site shape

All pause analysis focused on ChRO-seq data in three organs (heart, skeletal muscle, and kidney) which used a single base run-on of all four biotin nucleotides. We first focused our analysis on dREG sites in each tissue to identify regions enriched for transcription start and pause sites. We retained dREG sites in which we identified at least 5 reads mapping from both B6 and Cast alleles. We performed a K-S test to identify all candidate dREG sites that contained a candidate difference in pause, filtering for a false discovery rate of 0.1 (n=2784). To examine the relationship between initiation and pause, we identified the maxTSS and maxPause on the B6 and Cast allele separately using reads tagged with a SNP or indel. Since maxTSS and maxPause were defined independently, the maxPause was not always correctly paired with the maxTSS (*Tome et al., 2018*). We therefore used 2260 dREG sites where allelic maxPause were 10–50 bp downstream of allelic maxTSS on both alleles. These analyses pertain to *Figure 5A and B*.

### Identify genetic determinants of pausing

To focus on the genetic determinants of pausing that were independent of initiation, we identified changes in which the same maxTSS had different allelic maximal pause sites between the Cast and B6 alleles as follows. We used a K-S test to identify maxTSSs with a difference in the maxPause site between alles, filtering for maxTSSs with a different maxPause between alleles and a 10% FDR in a K-S test (n=269). In most analyses, we also draw a background set in which there was no evidence that sites sharing the same maxTSS had different maxPause sites between alleles, by identifying maxTSSs that have the same maxPause position and a K-S test FDR >0.9 (n=1396). In all analyses, we also filtered for maxTSSs with at least 5 allelic reads and B6/CAST read ratio between 0.5 and 2.

### Comparing GC content near the maxPause position

We compared the G, C, and GC content between alleles. We computed the G, C, and GC content as a function of position relative to maxPause. All of the G, C, and GC contents were combined across unique pause sites from all three organs for which we had single base resolution data (n=3456). We compared three blocks: block 1 was 11–20 bp upstream of maxPause, block 2 was 1–10 bp upstream of maxPause, and block 3 was 1–10 bp downstream of maxPause.

All scripts implementing analysis of allele-specific pause can be found at: https://github.com/Danko-Lab/F1_8Organs/tree/main/pause.

## Analysis of allele-specific termination

### Definition of allelic differences in termination

We noticed frequent AlleleHMM blocks near the 3' end of annotated genes. We used the transcription units (tunits) predictions which overlapped annotated protein coding genes (vM25), as these generally retained the window between the polyadenylation cleavage site and the transcription termination site. We identified transcription units that have AlleleHMM blocks starting within the transcription

unit and that end in the final 10% of the transcription unit or after the transcription unit. The overlapping region between the tunits and AlleleHMM blocks were called candidate allelic termination (AT) windows. To avoid obtaining candidate AT windows that reflected entire transcription units, we retain only AT windows whose length was less than or equal to 50% length of the host transcription unit.

## RNA-seq analysis
Allele-specific mapped RNA-seq reads using STAR (see above) were used as input to AlleleHMM to identify the region showing candidate allelic difference in mature mRNA. The transcription units that contain allelic termination windows, as defined above, were separated into two groups: One has an allelic difference in mature mRNA and the other does not. Those with an allelic difference in mature mRNA were defined as having the RNA-seq AlleleHMM blocks between 10 Kb upstream of the AT windows to the end of the AT windows.

## RNA stability analysis
We asked whether there was an allelic difference in mRNA stability between transcription units in which the allelic differences in termination affects the mature mRNA and those in which it does not. The RNA stability was defined as in *Blumberg et al., 2021*. The stability was defined as the ratio of RNA-seq read counts in exons to ChRO-seq read counts across the gene body. We used gene annotations from GENCODE (vM25). The RNA-seq reads were counted strand specifically using htseq-count. ChRO-seq reads were counted in a strand-specific fashion using in-house R scripts. After removing the genes with less than 10 B6-specific ChRO-seq and less than 10 CAST-specific ChRO-seq reads, the cumulative distribution functions were drawn. All differences were compared using a one-sided K-S test to compare differences in allelic RNA stability between groups.

All scripts implementing analysis of allele-specific termination can be found at: https://github.com/Danko-Lab/F1_8Organs/tree/main/termination.

## Acknowledgements
We thank M Garcia-Garcia, A Ozer, J Lis, H Kwak, G Barshad, A Chivu, and all members of the Danko lab for valuable discussions and suggestions throughout the life of this project. We thank P Borst and P Cohen for working with F1 hybrid mice and J Grenier and the Cornell TREx facility for preparing mRNA-seq libraries. We thank C Kaplan (U Pittsburgh) for rapid constructive comments based on our *bioRxiv* preprint. Work in this publication was supported by R01-HG010346 and R01-HG009309 (NHGRI) to CGD. The content is solely the responsibility of the authors and does not necessarily represent the official views of the US National Institutes of Health. Some of the figures in this manuscript were created using BioRender. All data are available at Gene Expression Omnibus under the accession number GSE174171.

## Additional information

### Funding

| Funder | Grant reference number | Author |
| --- | --- | --- |
| National Human Genome Research Institute | R01-HG010346 | Charles G Danko |
| National Human Genome Research Institute | R01-HG009309 | Charles G Danko |

The funders had no role in study design, data collection and interpretation, or the decision to submit the work for publication.

### Author contributions
Shao-Pei Chou, Formal analysis, Investigation, Methodology, Resources, Software, Validation, Visualization, Writing – original draft, Writing – review and editing; Adriana K Alexander, Investigation, Methodology, Resources; Edward J Rice, Methodology, Resources; Lauren A Choate, Formal analysis,

Methodology, Resources; Charles G Danko, Conceptualization, Funding acquisition, Project administration, Resources, Supervision, Writing – original draft, Writing – review and editing

**Author ORCIDs**
Lauren A Choate ⬤ http://orcid.org/0000-0003-4246-0550
Charles G Danko ⬤ http://orcid.org/0000-0002-1999-7125

**Ethics**
All mouse studies were conducted with prior approval by the Cornell Institutional Animal Care and Use Committee, under protocol 2004-0063.

**Decision letter and Author response**
Decision letter https://doi.org/10.7554/eLife.78458.sa1
Author response https://doi.org/10.7554/eLife.78458.sa2

## Additional files

**Supplementary files**
• Supplementary file 1. Table shows the number of reads mapped to B6 and CAST alleles in each cross and organ.

• Supplementary file 2. Table shows the coordinates (mm10) of each strain effect and imprinted domain. The organ column denotes which organs showed evidence of allele specific transcription.

• Supplementary file 3. Table shows the number of complete transcription units or initiation and pause windows with at least one genetic marker that can be used to assign allele specific transcription.

• Supplementary file 4. Table shows the design of all ChRO-seq adapter sequences used for each sample. Raw fastq file names are provided with each barcode. The JJ barcode column corresponds to an in-line barcode included in each 5' adapter. The i7 index sequence column corresponds to the Illumina barcode.

• MDAR checklist

**Data availability**
All data are available at Gene Expression Omnibus under the accession number GSE174171. All scripts are posted publicly with no restrictions on the Danko Lab GitHub organization, at: https://github.com/Danko-Lab/F1_8Organs (copy archived at swh:1:rev:8093a6c2ca1b15e869608c55bbef48dc539dad38).

The following dataset was generated:

| Author(s) | Year | Dataset title | Dataset URL | Database and Identifier |
|---|---|---|---|---|
| Danko CG, Chou SP | 2021 | Genetic dissection of the RNA polymerase II transcription cycle | https://github.com/Danko-Lab/F1_8Organs | GitHub, F1_8Organs |

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
