## [Editor Report]

This study exploits naturally occurring single nucleotide polymorphisms between mouse strains to provide novel information on how DNA sequence affects RNA polymerase II transcription initiation, termination, and pausing. The strength of this study lies in assessing how naturally occurring allele-specific polymorphisms impact the various steps in the transcription cycle in an unbiased and genome-wide manner.

---

## [Decision Letter]

**Decision letter after peer review:**

[Editors’ note: the authors submitted for reconsideration following the decision after peer review. What follows is the decision letter after the first round of review.]

Thank you for submitting the paper "Genetic Dissection of the RNA Polymerase II Transcription Cycle" for consideration by *eLife*. Your article has been reviewed by 3 peer reviewers, including Irwin Davidson as the Reviewing Editor and Reviewer #1, and the evaluation has been overseen by a Senior Editor.

After discussion, the referees were of the opinion that the current version of the study could not be considered for publication in *eLife*. A number of major issues need to be addressed that are described in the detailed comments of each referee below.

We would be willing to consider a novel version of the study that will be treated as a new submission if all of the issues can be satisfactorily addressed. However, we would like to emphasize the importance of a major overhaul of the presentation, terminology and clarity of the manuscript that all referees found was difficult to read and hence poorly accessible to non-specialist readers.

*Reviewer #1 (Recommendations for the authors):*

This study exploits naturally occurring single nucleotide polymorphisms (SNPs) between mouse strains to provide novel information how DNA sequence affects RNA polymerase II (Pol II) transcription initiation, termination and Pol II pausing. The study identifies nucleotide positions that impact these different processes, and the authors propose that functional effects of these changes may be explained by altering the energetic environments favorable for initiation or stability of Pol II pausing.

The strength of this study lies in assessing in an unbiased and genome wide manner how naturally occurring allele-specific SNPs impact the various steps in Pol II transcription. In contrast, as the study focuses mainly on a small window surrounding the initiation and pausing sites the authors did not assess SNPs that may affect the Pol II preinitiation complex and how these may also contribute to the observed effects.

The authors describe allele- and tissue-specific specific changes in imprinting. While this is of wide general interest, they have not addressed the mechanisms involved and thus its space devoted in the manuscript may not be appropriate. The manuscript could focus more on the aspects that have been analyzed in detail.

While the authors have analyzed the effects of SNPs on the major steps of the Pol II cycle, they do not provide detail on how these effects impact overall transcript levels. For example, do SNPs that affect pausing shifting Pol II to a less favored site lead to higher overall transcript levels? Can the authors indicate if any of the allele specific changes impact overall transcript levels?

The study focuses mainly on the SNPs located in a small window surrounding the initiation, pausing and termination sites. Can the authors comment on extending the study to SNPs that may affect binding of the Pol II preinitiation complex and how these may also contribute to the observed effects? First are there fewer SNPs at critical regions of the promoter aside the Inr element than at regions that show less consensus sequences? For example, are there frequent SNPs affecting TATA elements, or the TFIIB interacting regions, perhaps not the consensus per se but the flanking positions? Extending the study in this way may be important to provide independent evidence that the effects on a given SNP on start site or pausing are not further confounded by effects on PIC formation or positioning. This is important in the context of the authors model of SNPs providing or impacting energetically favorable environments.

*Reviewer #2 (Recommendations for the authors):*

Chou et al. present a "Genetic Dissection of the RNA Polymerase II Transcription Cycle" wherein they utilize reciprocal hybrid mouse crosses followed by transcriptomic analysis of specific tissues to determine cis effects on gene expression. In their analyses, they describe a cursory analysis of imprinted regions but focus more strain specific differences that are mappable in the hybrid. The three areas of focus are initiation position, pause position, and extended 3' transcribed regions that are specific to one allele vs the other. Plasticity in putative initiation position due to TSS mutations is especially interesting as this is not well studied in mammals and models remain to be tested for how promoters specify multiple start sites in a zone of initiation. The approach is quite interesting, and the work is carefully done, but there are a number of limitations. Descriptions of analyses are not always clear or accessible and presentation of some is not necessarily intuitive. A greater depth analysis of what fractions of genome are even accessible to the analysis with some clearer determination of what the extended 3' transcribed regions are, both those with and without changes to mRNA would improve the presentation. It is not clear from the presentation whether the strain-specific changes to mRNA associated with genes that have extended strain-specific 3' transcription are in composition of the mRNA or in expression.

Specific Points

The text and figures were perhaps less than straightforward to understand and there are a number of ways to potentially clarify presentation or extend analyses to be more compelling.

1. Abstract doesn't mention imprinting. While imprinting is not major focus, it is analyzed and discussed. To fully evaluate imprinting, a comparison of results here with prior over the subset of genome that would allow comparison would be valuable, especially given this work describes a subset of potentially novel imprinted transcripts.

2. Please be as clear as possible with terminology. There are a number of places where there is not a clear definition of terms (will be noted below). For differences in transcription termination, it is stated that these frequently do not change the composition of the mature mRNA, but the analysis that addresses what is meant by changes in composition is not clear. It appears to be strain bias in expression of the mRNA, and not instead alternative poly-adenylation or alternative splicing.

3. PDF p 4, 2nd para "On average, both strain-effect and imprinted domains spanned broad genomic regions (~10-1,000 kb; Figure 1G)"

It is difficult to imagine what a strain-effect domain actually entails in contrast to an imprinted domain where there is an idea of mechanism. If what is imagined is potential strain specific 3D genome organization or cis-regulation, potentially this could be made more clear. A more clear definition of what is meant by a domain- must it entail contiguous genes (I don't think it does)? It seems that a statistical analysis examining enrichment of strain-biased genes in certain regions of genome based on number of genes detectable by the analysis to begin with might be orthogonal approach to understand if close apposition is greater than chance (seems like it must be). No example of a strain-specific domain is actually shown and a deeper analysis of the characteristics of these domains would be useful.

4. How many genes used for correlation analysis? And how many genes have regions enabling detectability for strain specific expression? Could a gene-specific analysis be done and could it be shown in heat map across tissues? It seems that instead of just counting consistency across crosses and strain/imprinted would be visually apparent in such an analysis? This might be a way to give idea of the extent of effects relative to the size of the denominator- what genes are actually detectable in the strain-specific analysis.

5. Figure 1C is not as effective as I think it could be.

6. "TSN" and "TSS" are potentially confusing. TSS has been used in the literature to represent the idea of "TSN" while TSC (transcription start cluster) or TSR (transcription start region) have been used for how "TSS" is used here.

7. The analysis of initiation sites should be bolstered by a much greater level of comparison to bona fide TSS-seq or CAGE/other data for 5' ends. Given RNA fragmentation here and non-specific selection for capping, the authors are careful and they limit analysis to known TSS regions, but this analysis should be made more clear and a deeper comparison to either GRO-Cap or other data would help have confidence in the putative 5' ends analyzed.

8. To what extent do the changes in apparent TSS/TSR usage (in levels across TSRs) recapitulate the differentially expressed genes identified within the strain-specific domains (given these are distinct analyses). Making it clear that these are in fact different analyses- that there are some n (and what is the n) of TSS/TSRs that can be analyzed due to SNPs in the appropriate region at 5' end of gene vs some subset of genome/genes that is detectable in the AlleleHMM analysis (what fraction of genome? What number of genes?).

9. Figure 3B and other images of TSSs. SNP positions and sequence are meant or meant not to line up with the browser graph? SNPs should be made clear on what is present in each background- meaning annotate them different in way that is not confusing. If there are reads that can't be attributed to one or other background, these should be noted in the figure. Potentially a track for coverage that is indistinguishable?

10. When talking about "multiple" or "single" "base driven" allelic TSS difference, it appears what is meant is multiple TSN differences responsible for altered TSS usage in region vs say differences in TSS usage that might be associated with multiple SNPs. Because "base" is not clear, please find a way to make this as clear as possible.

11. For Pol II scanning, please add references to Giardina and Lis (10.1126/science.8342041), and Kuehner and Brow (10.1074/jbc.M601937200).

12. PDF p. 12 "Candidate initiator motifs within 20 bp of the CA/ non-CA initiation site had slightly more initiation signal on the non-CA allele compared with the CA allele, consistent with the shooting gallery model but inconsistent with the prevailing model of independent TSNs expected in mammals (Figure 4B; purple). By contrast, TSNs where both alleles contained a CA dinucleotide (n = 8,147 [brain] and 10,113 [liver]) did not show this same effect (Figure 4B; gray)."

If usage of YR elements in mouse examined, I think there is a hierarchy -> CA>TA{greater than or equal to}TG>CG. The analysis could be done differently to reflect this (CG being by far the most disfavored but also the most prevalent in TSS/TSR regions). What about consensus initiators (more information than just YR)- do they behave differently or the same? Is there a more sophisticated analysis that instead looks at fraction of distribution that is lost from max TSN vs fraction gained in next YR based on identity of that YR? Or simply whether the distribution in the TSS/TSR shifts upstream or downstream on average? With the sample size differences between affected and unaffected, would bootstrapping of the unaffected be a useful way to detect significant difference of the affected sample? In text the caveat of multiple SNPs having differential effects on potentially overlapping PICs should be made stronger- for example affecting an upstream site through DPE etc.

13. Figure 4C "The box plots show the distribution of ChRO-seq signals ratios at TSNs with any variation of the initiator motif (including CA, CG, TA, TG)"

This sentence could be interpreted as "variation in the initiator motif" meaning SNPs in the initiator motif, when what is meant is all four YR sequences were considered.

14. Figure 4D. The sequence relative to the background should be made clear. Potentially just label both B6/CAST positions that are different, especially because it is not clear that the B6 color code sequence actually lines up given how the genome browser is displaying adjacent positions.

15. PDF p. 16 "Changes in pause position were correlated with changes in the size of the insertion or deletion, such that the distance between the max TSN and the pause site in the native genome coordinates was typically less than 5 bp (Figure 5C)."

Is it meant here that there appears to be a defined pause distance and therefore with INDELs pause site moves relative to genome, but stays consistent in number of bases between TSN and pause? Otherwise, I am having a hard time understanding what is meant here.

16. PDF p. 16 "Allelic differences at the C base had the strongest association with the pause, followed by the -2 and -3 G bases"

This is unclear. It appears what is meant is that SNPs at 0, -2, -3 are most enriched for pauses sites with pause differences- referral to bases C,G is confusing because these are not present at all pauses, just enriched across pauses? Whatever is meant here, please make more clear.

17. "These observations support a model in which DNA sequence changes that disfavor pausing result in Pol II slipping downstream to the next pause site for which DNA sequence is energetically favorable, but maintaining physical connections that may exist between paused Pol II and the PIC."

It is unclear what is meant by "maintaining physical connections between the pause and the PIC"

18. Figure 6A. What about any downstream changes associated with a AT window or adjacent to it? Are the AT events enriched inside larger strain-specific domains? How do these relate to other aspects of altered expression in region. In the figure example there does look like there is a potential CAST leaning Liver dREG in the figure. How does the height of the peaks on the "all" browser track relate to the Cast and B6, it does not look like sum of the two plus reads that can't be distinguished? Is it instead only reads that can't be distinguished? Is this different from what is show for TSN/TSS browser views?

19. For considerations about association of energetic favorability of difference sequences, especially as relates to pausing, a critical caveat is any bias in sequence that favors detection by ChRO-seq, which will only be a subset of elongation complexes.

20. PDF p. 30 "Comparison of AT consent between alleles" -> typo

21. PDF p. 31 "One has an allelic difference in mature mRNA and the other does not. Those with an allelic difference in mature mRNA were defined as having the RNA-seq AlleleHMM blocks between 10Kb upstream of the AT windows to the end of the AT windows."

I suspect the majority of reader would read allelic difference in mRNA composition would relate to altered mRNA content, either APA or alternative splicing, but here it appears to relate to strain-specific expression of the RNA that shows AT, but another reading is that this analysis detects APA via differential expression over region where 3'UTRs are different. Please make this more clear- it also could easily be shown that over gene itself – where detectable – there are not allelic differences and therefore allelic differences in 3' UTR or identification of allele-specific splicing really suggest altered mRNA processing.

22. Supplemental Figure 2C. "at sites showing allelic differences in TSSs driven by a single base in the brain." base -> TSN to be more clear.

*Reviewer #3 (Recommendations for the authors):*

This manuscript from the Danko lab describes nascent RNA-sequencing of eight tissues collected from heterozygous F1 mice. Using the known sequences of each parental allele, nascent RNA was analyzed to define regions of strain effect, meaning there was a significant difference in sites of transcription initiation, transcriptional pausing, or termination depending on the strain. The authors then investigated genetic variation between parental alleles to shed light on specific nucleotides or sequence contexts of functional relevance. A large variety of different aspects of transcription are touched on, none in great depth, and the findings are generally in agreement with earlier work using alternative methods. Due to the low number of sites detected where differences between strains were apparent, the study is somewhat underpowered to make strong or new conclusions. In addition, it was not discussed whether small (5-10bp) shifts in sites of transcription initiation or pausing would have relevance for gene expression, making it unclear how to connect these findings to a biological effect.

Strengths of this study include the clever study design, and sophisticated computational approach. This work could perhaps best be characterized as a methods paper, demonstrating a number of possible ways that F1 hybrid mice can be used to study gene regulation. In this regard, the authors succeed nicely.

Weaknesses of the work derive from the small number of events detected wherein sequence differences between the parental strains appear to alter transcription initiation, pausing or termination. As a result, this work isn't able to tease out new mechanisms or bring us closer to being able to predict how sequence changes would affect transcription levels, but instead confirms or supports existing models.

Results concerning the mechanism of search by RNA polymerase II for a site to initiate have more promise to reveal new findings. It would be interesting to provide compelling evidence of Pol II scanning in mammals, as it is known to do in yeast. The analysis as presented seems underpowered by a small number of instances. Perhaps the authors could dig into their data for other tissues or aggregate across tissues to extract more from this analysis?

Finally, results are presented indicating that differences in locations of transcription termination can be detected between the strains. This is the one instance where the authors probe steady state RNA-seq to determine if such differences affect levels of mature mRNA. In this case, the answer appears to be generally 'no', suggesting that the cleavage and polyadenylation site is the same between strains, and it is only the location of Pol II termination after mRNA is cleaved and released that differs between alleles. Thus, this finding, as interesting as it may be, does not appear to have a biological consequence.

Overall:

I was impressed with the approach but underwhelmed with the resulting findings.

I find the analyses to be lacking in depth and presented more as a hodge-podge of vignettes rather than a connected story. As a result, there are some potentially interesting nuggets of information uncovered, but these are often presented without biological context or interpretation. As a result, I wasn't sure what the 'take-home' messages of the work might be.

For example, results concerning transcription initiation are solid, but unsurprising: mutations in the core of the Initiator motif (e.g., the A residue with the highest information content and weight in a PWM) are the most important for determining levels of initiation at that location, and the AT-richness of the transcription bubble and energetics of melting play some role in choice of initiation site. Nice findings, and rigorous work, but the authors might need to collect more data or dig deeper into the analyses to break new ground.

Specific concerns are:

1) the authors state that strain-effect domains that showed allelic imbalance 'were generally composed of multiple transcription units.' however, figure 1E shows that >40% of the strain effect domains contain only 1 annotated gene, and ~30% contain 2 genes. If the figure is accurate, and >70% of the strain effect domains contain 1-2 genes, I don't see how this is in agreement with the text. The percentage of imprinted domains with >1 gene seems higher, but with only 28 domains to study, it is unlikely that one can draw strong conclusions from this.

2) I will note that I was struck by how few imprinted domains were observed. Do the authors think that this results from the short read length of Chroseq, which makes it rare to have a strain specific SNP in a read? If so, have the authors considered using RNA-seq to define differences in RNA levels between strains, and then to use Chroseq to investigate what steps in the transcription cycle and which SNPs might underlie these effects? Naively, it seems like including a method like RNA-seq with longer reads and the ability to determine whether changes are biologically meaningful could strengthen the work and allow more interesting or novel conclusions to be drawn.

3) Presentation of some results could be improved to increase clarity. In particular, I was interested in the data shown in Figure 4, but struggled for some time to truly understand exactly what was being shown. I worry that the presentation style will limit the readership.

[Editors' note: further revisions were suggested prior to acceptance, as described below.]

Thank you for resubmitting your work entitled "Genetic Dissection of the RNA Polymerase II Transcription Cycle" for further consideration by *eLife*. Your revised article has been evaluated by Kevin Struhl (Senior Editor) and a Reviewing Editor.

The manuscript has been improved but there are some remaining issues that need to be addressed, as outlined below:

This new version of the manuscript has addressed many of the issues raised by the referees on the original version. It is more focused and concise. The referees nevertheless feel that while the paper will be of wide general interest to readers at *eLife*, a further revision of the manuscript is required. A number of issues listed below need to be addressed. We would draw your attention in particular to the comments concerning other approaches for evaluating TSS distribution as alternatives to the Kolmogorov-Smirnov test and the questions concerning the new data on the TTS and Poly A site usage. The authors should also pay attention to the suggested changes that may enhance accessibility of the paper to non-specialist readers.

*Reviewer #1 (Recommendations for the authors):*

This new version of the manuscript has addressed many of the issues raised by the referees on the original version. It is more focused and concise. The addition of new data has raised several issues that need to be addressed.

The authors have added new data concerning how SNPs affect Pol II transcription termination. The data show that many SNPs lead to longer primary transcripts raising the possibility of use of alternative polyadenylation sites. To address this, the authors sequence the corresponding PolyA+ transcripts and claim that a significant fraction show an altered primary structure, in some cases with inclusion of additional exons. This section is very confusing as the authors make no mention of use of alternative polyadenylation sites. They are not mapped or discussed. Also, it is not clear why having a longer primary transcript would alter the use of the original polyadenylation site. While 21-41% of transcription units have altered primary mRNA structure, does this mean that the major polyadenylation site is no longer used or used at a lower frequency. Which % of the total transcripts from a given gene actually show the altered primary structure. This issue also is neither mentioned nor discussed.

*Reviewer #2 (Recommendations for the authors):*

The manuscript is improved and more focused. My concerns mostly relate to accessibility of results. Throughout the results the manuscript could be improved if framework for understanding results were more clear. Explicitly stating possibilities in each case and then indicate what types of changes the analyses detect and what they suggest.

Abstract

1. "Our results implicate the strength of base pairing between A-T or G-C dinucleotides as key determinants to the position of Pol II initiation and pause."

This is maybe a little strong for the abstract.

Results

2. "Our results suggest a hierarchy of initiation frequency, in which Pol II prefers to initiate at a CA dinucleotide, followed by TA, TG, and finally CG"

I believe this also fits the observed usage hierarchy observed in TSS-seq in humans/mice (potentially check if there are appropriate references for this but note that usage should be normalized to dinucleotide frequency). Also note, it is different than yeast preference https://doi.org/10.1101/2021.11.09.467992

3. "We therefore developed a statistical approach based on the Kolmogorov-Smirnov test to identify differences in the shape of the distribution of Pol II initiation (see Methods)."

This potentially seems fine but note that there are other definitions of shape and tools used to determine differences in index. I suspect the original "shape" index might not necessarily work for more subtle changes (see https://genome.cshlp.org/content/21/2/182.long) but that might be what readers might think shape of TSS distribution is referring to. For potentially alternative framework or useful statistical approach to examining differences in distributions see: https://academic.oup.com/nargab/article/3/2/lqab051/6290625

4. Section starting with sentence "Next we examined the distribution of SNPs centered on allele specific max TSSs"

I think these paragraphs could be more clear if possibilities for differences were imagined for the reader and then the analyses introduced. What is being done here is asking where SNPs are relative to the affected site when the change between alleles appears to be driven by differences in a single site or multiple sites. And the answer is the SNPs tend to be right on top of the affected TSS.

5. Section just below above, paragraph starting with "Intriguingly, although the increased density of SNPs observed was largest near the Inr element, SNPs were enriched throughout the region occupied by the PIC in both single and multiple TSS driven allele specific TSCs (Figure 3C,E)."

This section is relatively dense and not really accessible. This section appears to be asking whether SNPs might be affecting allele-specific expression through changes in and around TATA. The answer appears to be yes, to some extent. I think the paragraph could be much more clear in that regard. [Please also review the section on pausing to ask how accessible it is]

6. Section starting "Models of stochastic search during transcription initiation". If term "shooting gallery" is used to describe initiation by promoter scanning, it should be explained briefly ie. "a directional process where rate of catalysis and processivity and rate of DNA translocation determine distribution of TSS usage" and stressing that upstream TSSs will by default have priority over downstream ones though still reflecting innate Inr strengths.

7. "This indicates that changes in allelic termination often had a corresponding impact on the primary structure of the mature mRNA."

"impact" seems to suggest causality so potentially use a different word? I think you are just saying there is an apparent correlation when IDing change in termination with IDing altered primary structure (though potentially should also do the analysis in reverse, if possible).

Figures

8. Please check all figures and legends. PDF conversion appears to have corrupted spacing on many graph legends, but additionally, there are a few "muntants" and a "browian".

9. Figure 2-sup 1

"(B) Scatter plot shows the number of TSSs in a TSC as a function of the read counts in the

TSC."

Consider analyses but only counting sites with some threshold of total usage, e.g. 1% or 2%- this might give a better idea about real vs just seeing something at a position because read depth increases.

Figure 6-sup 1 A and B legends are switched.

*Reviewer #3 (Recommendations for the authors):*

In this revised submission, the authors have addressed the major concerns of previous reviewers and, as a result, produced a much revised but stronger and more focused manuscript. The manuscript covers an important topic, the genetic determinants of RNA polymerase II activity using an excellent and well designed system, the F1 mouse. Below I outline my one concern which should still be addressed.

Much of their work on the shape of transcription initiation hinges on the Kolmogorov-Smirnov test, which -- while statistically valid, is quite permissive in calling two things different. Given the high number of initiation regions with KS-test differences, the fact that many of these do not overlap strain effect domains, and the fact that replicates in nascent protocols can have subtle shape differences by random chance -- I have to wonder how many false calls they would expect using this KS approach. As they condition on B6 vs CAST alleles for the KS-test, one possibility would be to apply the test within strain but across replicates to estimate a false positive rate -- though admittedly data amount may be limiting in this scenario.

---

## [Author Response]

[Editors’ note: the authors resubmitted a revised version of the paper for consideration. What follows is the authors’ response to the first round of review.]

Reviewer #1 (Recommendations for the authors):This study exploits naturally occurring single nucleotide polymorphisms (SNPs) between mouse strains to provide novel information how DNA sequence affects RNA polymerase II (Pol II) transcription initiation, termination and Pol II pausing. The study identifies nucleotide positions that impact these different processes, and the authors propose that functional effects of these changes may be explained by altering the energetic environments favorable for initiation or stability of Pol II pausing.The strength of this study lies in assessing in an unbiased and genome wide manner how naturally occurring allele-specific SNPs impact the various steps in Pol II transcription. In contrast, as the study focuses mainly on a small window surrounding the initiation and pausing sites the authors did not assess SNPs that may affect the Pol II preinitiation complex and how these may also contribute to the observed effects.The authors describe allele- and tissue-specific specific changes in imprinting. While this is of wide general interest, they have not addressed the mechanisms involved and thus its space devoted in the manuscript may not be appropriate. The manuscript could focus more on the aspects that have been analyzed in detail.

We fully agree with this comment. We have removed two of the sections on imprinting in the brain and the section describing imprinted lincRNAs from the revised manuscript, as well as an associated figure (previously Figure 2). We still show some information on imprinting in Figure 1, which we think is important for validating new ChRO-seq data prepared for this manuscript and to set the stage for our in-depth analysis of the Pol II transcription cycle. As a result, we believe that the revised manuscript is a much more cohesive story than our original submission.

While the authors have analyzed the effects of SNPs on the major steps of the Pol II cycle, they do not provide detail on how these effects impact overall transcript levels. For example, do SNPs that affect pausing shifting Pol II to a less favored site lead to higher overall transcript levels? Can the authors indicate if any of the allele specific changes impact overall transcript levels?

We have conducted new analyses comparing allelic changes in the Pol II transcription cycle to allelic differences in gene body transcription and mRNA.

In some cases, we do see a clear effect on mRNA primary structure and transcript levels. One clear example is allelic changes in the site of Pol II termination, for which we have found a clear impact on mRNA primary structure (i.e., the set of all nucleotides in a transcript), mRNA stability, and transcript level. In our revision, we conducted a new analysis which found that allelic changes in mRNA primary structure are up to 4-fold enriched in genes with allelic differences in Pol II termination. These changes in primary structure were also associated with differences in the stability of the processed mRNAs, which we determined by comparing the ratio of mRNA to transcription (*p* = 2.7e-8 [liver]; *p* = 4e-4 [brain]; Kolmogorov-Smirnov test). Finally, these changes were associated with allelic differences in the mRNA levels (*p* = 1.6e-5 [liver]; p = 0.015 [brian]; Wilcoxon rank sum test combined with Fisher’s method). These results are all reported in the revised manuscript, in particular in the section titled “Allelic changes in termination correlate with differences in mRNA stability”, and in Figure 6 and Figure 6—figure supplement 1 and 2.

We also examined the correlation between transcription initiation and mRNA during our revision. Overall, changes in the levels of transcription initiation in transcription start clusters (TSCs) were 3-5-fold enriched in strain effect domains, indicating that these differences in transcription were, on average, associated with differences in Pol II abundance on gene bodies.

We report this association in the revised manuscript (see, in particular, the 3rd paragraph in the section “Widespread genetic changes in transcription initiation”).

Throughout the initiation and pause sections, most of our analysis focused on changes in the precise location of initiation or pause. Genome-wide these changes are not strongly enriched in changes in mRNA. Our intuition is that natural selection acting to preserve the protein function allows changes in the precise coordinates of initiation or pause at many genes, leading to slight differences in the mRNA with little to no phenotypic impact. These findings are also conceptually consistent with a recent preprint from the Andersson lab (Einarsson et al. (2021) bioRxiv).

Despite not having a strong impact on mRNA abundance, shifts in the precise position of Pol II provide a window into transcriptional mechanisms that are of interest. To emphasize these points in the revised manuscript, we have rewritten the rationale for many of the analyses shown in the revised paper. Additionally, we have made substantial changes that are designed to emphasize the findings with the most exciting implications, such as our model of brownian motion following Pol II initiation. These changes are too numerous to list exhaustively, but include (for instance) substantial changes and new analysis described in the section titled “Models of stochastic search during transcription initiation”.

The study focuses mainly on the SNPs located in a small window surrounding the initiation, pausing and termination sites. Can the authors comment on extending the study to SNPs that may affect binding of the Pol II preinitiation complex and how these may also contribute to the observed effects? First are there fewer SNPs at critical regions of the promoter aside the Inr element than at regions that show less consensus sequences? For example, are there frequent SNPs affecting TATA elements, or the TFIIB interacting regions, perhaps not the consensus per se but the flanking positions?

We have examined the distribution of SNPs in DNA sequences that bind to general transcription factors in the pre-initiation complex.

First, we have added the position of TATA-box and BRE containing regions to the SNP density maps in Figure 3 C and E and Figure 3—figure supplement 1 B and D. We do observe a global enrichment of the density of SNPs within the window that binds the pre-initiation complex, although the enrichment is a lower magnitude than observed over the initiator element. We did not observe any consistent evidence for an increased density of SNPs located in the windows overlapping the classical positions of TATA (between -31 and -24) or BRE (between -32 and -37) compared with surrounding DNA. Rather, SNPs are enriched throughout the region indicated relative to control. This finding may be consistent with the reviewer’s proposal that sequence flanking core motifs play a role.

To explore the impact of PIC motifs further, we focused on the TATA box. As the TATA box only occurs in a relatively small percentage of mammalian promoters (Sandelin et al. (2007) Nature Reviews Genetics), we also conditioned on the presence of a clear TATA-like motif on at least one of the alleles. We first used a TATA motif that had a general enrichment for AT content, consistent with the degenerate nature of the TATA box in mammals (see Methods). We found that SNPs which changed the TATA motif had a low, positive correlation with allele specificity, with a slightly stronger correlation in brain than liver (Pearson’s R = 0.09 [liver]; R = 0.18 [brain]). The positive correlation was marginally significant in the brain (*p* = 0.04), but not in the liver (*p* = 0.2). Similar results were also obtained with an additional TATA motif that was a stronger match to the classical TATA consensus (TATAAA; p = 0.078 [liver]; p = 0.051 [brian]). We interpret these results to be consistent with TATA being an important, but fairly weak, determinant of the location of transcription initiation in a subset of promoters in mammals. These results are now reported in the revised manuscript, see in particular paragraph 5 in the section “Allelic changes in the shape of transcription initiation”.

Extending the study in this way may be important to provide independent evidence that the effects on a given SNP on start site or pausing are not further confounded by effects on PIC formation or positioning. This is important in the context of the authors model of SNPs providing or impacting energetically favorable environments.

Our findings above suggest that positions harboring TATA and BRE, which impact PIC formation and positioning, are less enriched for SNPs than positions directly adjacent to the TSS. We also wish to note that when we analyze changes in the shape of paused Pol II, we limit our analysis to reads that have the same TSS, requiring that PIC positioning effects on pause do not alter the TSS.

We do acknowledge the caveat that some of the SNPs overlapping the TSS may affect similarity of that sequence to TATA, BRE, or other PIC motifs (also pointed out by reviewer 2 comment 12). Although we think this explanation is less likely than the direct impact on the initiator element (as our intuition is that such changes are unlikely to cluster on or after the TSS or pause site), we do nevertheless think this is an important caveat to note in our discussion. We have addressed this point in the revised discussion, where we write:

“Another possible interpretation is that changes in the CA dinucleotide sequence alter DNA elements that support partially overlapping PICs (e.g., CA -> TA dinucleotide change makes the sequence slightly closer to a TATA box that could support initiation further downstream). Although we do not directly rule out this alternative interpretation, in our view it seems less likely because the effects we observe are constrained within a fairly narrow window, because the effects we observed are so consistent when conditioning on changes in the initiator DNA sequence motif, and because DNA sequence changes in the TATA box did not have as large of an impact on the initiation site. In either case, however, it is clear from our data that changes in the DNA sequence of initiator elements tend to increase the use of candidate initiators nearby.”

Reviewer #2 (Recommendations for the authors):Chou et al. present a "Genetic Dissection of the RNA Polymerase II Transcription Cycle" wherein they utilize reciprocal hybrid mouse crosses followed by transcriptomic analysis of specific tissues to determine cis effects on gene expression. In their analyses, they describe a cursory analysis of imprinted regions but focus more strain specific differences that are mappable in the hybrid. The three areas of focus are initiation position, pause position, and extended 3' transcribed regions that are specific to one allele vs the other. Plasticity in putative initiation position due to TSS mutations is especially interesting as this is not well studied in mammals and models remain to be tested for how promoters specify multiple start sites in a zone of initiation. The approach is quite interesting, and the work is carefully done, but there are a number of limitations. Descriptions of analyses are not always clear or accessible and presentation of some is not necessarily intuitive. A greater depth analysis of what fractions of genome are even accessible to the analysis with some clearer determination of what the extended 3' transcribed regions are, both those with and without changes to mRNA would improve the presentation. It is not clear from the presentation whether the strain-specific changes to mRNA associated with genes that have extended strain-specific 3' transcription are in composition of the mRNA or in expression.Specific PointsThe text and figures were perhaps less than straightforward to understand and there are a number of ways to potentially clarify presentation or extend analyses to be more compelling.1. Abstract doesn't mention imprinting. While imprinting is not major focus, it is analyzed and discussed. To fully evaluate imprinting, a comparison of results here with prior over the subset of genome that would allow comparison would be valuable, especially given this work describes a subset of potentially novel imprinted transcripts.

At the suggestion of Reviewer #1, we have removed the imprinting section. We agree it does not fit well with the remainder of the paper. Removing imprinting from this manuscript provides more space for us to flesh out some of the more interesting observations about the Pol II transcription cycle.

2. Please be as clear as possible with terminology. There are a number of places where there is not a clear definition of terms (will be noted below). For differences in transcription termination, it is stated that these frequently do not change the composition of the mature mRNA, but the analysis that addresses what is meant by changes in composition is not clear. It appears to be strain bias in expression of the mRNA, and not instead alternative poly-adenylation or alternative splicing.

We fully agree with the reviewer. We have attempted to clarify our terminology throughout the revised manuscript (as noted in response to specific points, below).

To clarify the mRNA section pointed out by the reviewer, we actually examined both the mRNA primary structure (i.e., the alternative use of blocks of ribonucleotides in the mRNA between alleles), mRNA expression (i.e., the number of reads), and the relationship between primary structure and expression. We have re-written the main text to clarify what comparison we are making in each section. In particular, we now use the term “primary structure” to describe changes in the composition of the mature mRNA. To emphasize this point, we have also included examples of changes in primary structure in the supplementary figures. Finally, we have moved a discussion of mRNA expression to the final paragraph, and added a topic sentence so that readers are aware of the traisition. We believe it is much more clear which changes we are examining in each section of the revised manuscript.

3. PDF p 4, 2nd para "On average, both strain-effect and imprinted domains spanned broad genomic regions (~10-1,000 kb; Figure 1G)"It is difficult to imagine what a strain-effect domain actually entails in contrast to an imprinted domain where there is an idea of mechanism. If what is imagined is potential strain specific 3D genome organization or cis-regulation, potentially this could be made more clear. A more clear definition of what is meant by a domain- must it entail contiguous genes (I don't think it does)? It seems that a statistical analysis examining enrichment of strain-biased genes in certain regions of genome based on number of genes detectable by the analysis to begin with might be orthogonal approach to understand if close apposition is greater than chance (seems like it must be). No example of a strain-specific domain is actually shown and a deeper analysis of the characteristics of these domains would be useful.

We have rewritten the Results section to lead into the idea of strain effect domains in a more cohesive and hypothesis driven manner. New text is backed by new analysis and examples, as well as a more clearly defined biological explanation for strain effect or imprinted domains. Specific changes to the manuscript include the following:

– We directly tested the hypothesis that pairs of nearby transcripts change more frequently than expected by chance. In this test, we compared two groups of transcript: One that has significant difference in Pol II abundance between alleles and one that does not. We found that for transcripts which have a significant change in allele specificity, the adjacent gene tends to have a difference in allele specificity as well (Fisher's Exact Test, BN: p-value = 0.0005348, LV:p-value < 2.2e-16). We think this new analysis supports the reviewers’ intuition that close apposition of genes that have evidence of a strain effect is higher than chance. This new analysis is described in the Results section (see the section Atlas of allele specific transcription in F1 hybrid murine organs, paragraph 3).

– We show an example of a strain effect domain in the revised Figure 1—figure supplement 1D. This provides an example of allelic differences in the transcription abundance of multiple transcripts located near one another along the chromosome.

– We have re-written the manuscript to more clearly articulate the types of mechanisms that could influence multiple transcripts located near one another. We specifically hypothesize that clustered changes are “consistent with differences affecting regulatory regions that control the activity of broad transcription domains”. These may reflect locus control regions, enhancers affecting multiple transcripts in a locus, or the effects on lincRNAs that function in an activating or repressive manner, similar to those impacted in imprinting. We think our observations reflect similar underlying biology as those which were based on cross-individual variation or expression noise in different biological replicates (Delaneau et al., 2019; Kumasaka et al., 2019; Rennie et al., 2018).

4. How many genes used for correlation analysis? And how many genes have regions enabling detectability for strain specific expression? Could a gene-specific analysis be done and could it be shown in heat map across tissues? It seems that instead of just counting consistency across crosses and strain/imprinted would be visually apparent in such an analysis? This might be a way to give idea of the extent of effects relative to the size of the denominator- what genes are actually detectable in the strain-specific analysis.

The revised manuscript includes a new main figure panel that shows allelic differences in transcription between annotated genes similar to the one which we think was suggested by the reviewer (see the revised Figure 1C). We think this visualization helps to show readers that the majority of allelic differences arise in a single organ.

To address the question about the fraction of genes that can be analyzed for strain specific expression, we have included a new supplementary table which shows statistics for the proportion of transcripts that could be identified as differentially expressed between alleles based on the presence of sufficient genetic markers to assign allele specificity (see Supplementary Table 3). Briefly, Supplementary Table 3 shows that the vast majority of long genes can be analyzed for allele specific expression without any difficulty, though the proportion of TSSs and pause sites marked by a SNP is lower because fewer tagged SNPs are found across the much shorter window in these regions.

Additionally, to help readers interpret the meaning of numbers of changes, we provide the fraction of changes from the universe of tagged examples throughout the revised Results section. Changes are too extensive to list in detail, but some include:

See the section "Atlas of allele specific transcription in F1 hybrid murine organs"

“We identified 1,374 genes and lincRNAs with strong evidence that transcription was significantly higher across the gene body on either the B6 or CAST allele in at least one organ, comprising about 8% of the 17,703 annotated genes.”

And for transcription initiation, see the section "Widespread genetic changes in transcription initiation"*.*

“Allelic changes in TSSs were common, occuring in ~16-34% of TSCs tagged with SNPs (n = 1,109 – 5,793; binomial test FDR < 0.1; Supplementary Table 3).”

5. Figure 1C is not as effective as I think it could be.

We agree with the reviewer that the cartoon previously shown in Figure 1C does not add enough clarity to be effective as-is. We have made three changes in response. First, we have moved the panel that was previously Figure 1C to the supplement (Figure 1—figure supplement 1E). Second, we have re-written the Results section to help motivate the rationale for examining broad domains (see paragraph 3 in the section "Atlas of allele specific transcription in F1 hybrid murine organs"). We think these writing changes help by clarifying what we intend to identify using this procedure in a biological context. Third, in the revised Figure 1—figure supplement 1E (previously Figure 1C) we have simplified the description of each subpanel. Our goal is to complement the detailed description of heuristics we used to group AlleleHMM blocks into domains in the *Methods* section. We think these changes help make the manuscript more accessible to readers.

6. "TSN" and "TSS" are potentially confusing. TSS has been used in the literature to represent the idea of "TSN" while TSC (transcription start cluster) or TSR (transcription start region) have been used for how "TSS" is used here.

We agree with the points raised by the reviewer. We have changed the language used throughout the manuscript. TSS now represents individual nucleotides and TSC represents clusters of TSSs, and TSR represents regions with multiple transcription start sites. These terms are defined in the revised Figure 3A.

7. The analysis of initiation sites should be bolstered by a much greater level of comparison to bona fide TSS-seq or CAGE/other data for 5' ends. Given RNA fragmentation here and non-specific selection for capping, the authors are careful and they limit analysis to known TSS regions, but this analysis should be made more clear and a deeper comparison to either GRO-Cap or other data would help have confidence in the putative 5' ends analyzed.

We fully agree with this suggestion. We used our pipeline to analyze paired-end PROseq data in K562 cells, where PRO-cap data exists for comparison. We found that candidate max TSSs identified using our approach have more PRO-cap signal at that individual nucleotide than using human gene annotations as a negative control (Figure 2—figure supplement 2C). Our result indicates that we are, at least on average, identifying the location of TSSs with significant cap signal. This validation is written into the revised Results section and complements the discovery of the initiator motif.

8. To what extent do the changes in apparent TSS/TSR usage (in levels across TSRs) recapitulate the differentially expressed genes identified within the strain-specific domains (given these are distinct analyses).

We have examined the enrichment of allelic differences in TSSs in strain-specific domains in our revised manuscript. Changes in the total abundance of Pol II were highly enriched in changes in AlleleHMM blocks (Fisher's Exact Test, BN p-value < 2.2e-16, odds ratio 5.375217 ; LV p-value = p-value < 2.2e-16, odds ratio 3.474105). This is now noted in our revised manuscript, see especially the 2nd paragraph in the section “Widespread genetic changes in transcription initiation”, where we write:

“Allelic changes in TSSs were common, occurring in ~16-34% of TSCs tagged with SNPs (n = 1,109 – 5,793; binomial test FDR < 0.1; Supplementary Table 3). Changes in TSSs were highly enriched within strain effect domains identified by AlleleHMM (Chou and Danko, 2019) (odds ratios 3.5-5.4; p < 2.2e-16, Fisher’s exact test). Therefore, many of these allelic changes in TSSs likely reflect allelic changes in the rates of transcription initiation on the gene secondary to allelic differences in transcription factor binding or other regulatory processes. These mechanisms have been explored extensively elsewhere (Battle et al., 2014; Chen et al., 2016; Lappalainen et al., 2013; Montgomery et al., 2010; Pickrell et al., 2010).”

In addition to changes in initiation levels across TSRs, we have also conducted an analysis on the “shape” of initiation signal inside TSCs. These changes in initiation shape do not overlap strain effect domains as well as those identified using TSS levels (only 10-20% of the TSCs with changes in shape were also found inside of strain effect domains). This is expected because compensatory changes between alleles, which change the shape of initiation signal in TSCs, can provide consistent amounts of initiation at genes with selection.

We have rewritten the Results section to clarify that these two types of changes are separate and are both interesting for different reasons. The changes to the text are too extensive to list completely, but involve a large rewrite of the sections "Widespread genetic changes in transcription initiation and Allelic changes in the shape of transcription initiation".

Making it clear that these are in fact different analyses- that there are some n (and what is the n) of TSS/TSRs that can be analyzed due to SNPs in the appropriate region at 5' end of gene vs some subset of genome/genes that is detectable in the AlleleHMM analysis (what fraction of genome? What number of genes?).

We provide a table (Supplementary Table 3) in the revised manuscript which shows the number of TSSs, pause sites, and transcription units that have enough genetic markers between CAST and B6 strains to analyze allelic differences in transcription. Briefly, for transcription units we have enough markers to reliably estimate allele specific transcription in >~90% of transcription units. As expected, the fraction of transcription start clusters that could be analyzed for changes in initiation or pause sites is much lower (~20%), because the SNP needs to be found in a much more precise location. In addition to the table showing the numbers of each functional type that could be analyzed for changes between alleles, we also use this number as the denominator when reporting the fraction of genes, TSS/ TSC/ TSR, and pause sites that change between alleles throughout the revised manuscript. For instance, see the 2nd paragraph in the section “Widespread genetic changes in transcription initiation”, where we write:

“Allelic changes in transcription initiation frequency were common, occuring in ~16-34% of TSCs tagged with SNPs (n = 1,109 – 5,793; binomial test FDR < 0.1; Supplementary Table 3).”

9. Figure 3B and other images of TSSs. SNP positions and sequence are meant or meant not to line up with the browser graph? SNPs should be made clear on what is present in each background- meaning annotate them different in way that is not confusing. If there are reads that can't be attributed to one or other background, these should be noted in the figure. Potentially a track for coverage that is indistinguishable?

We have clarified the presentation of markers in the revised manuscript. We now indicate the nucleotides for B6 and SNPs in CAST in separate rows. All DNA sequence tracks that show DNA sequence are color coded to indicate the nucleotide at each position. We have also attempted to fix any alignment issues between ChRO-seq and DNA sequence.

Additionally, we now include a 4th track which shows reads that cannot be assigned to a particular allele because they are not tagged with SNPs for all examples of allelic changes in initiation position in Figure 3B. We agree that this change complements the existing “all reads” track and makes it easier to interpret this figure.

10. When talking about "multiple" or "single" "base driven" allelic TSS difference, it appears what is meant is multiple TSN differences responsible for altered TSS usage in region vs say differences in TSS usage that might be associated with multiple SNPs. Because "base" is not clear, please find a way to make this as clear as possible.

The reviewer is correct – “multiple base driven” changes reflect differences in multiple TSSs (rather than changes tagged by multiple SNPs). To clarify this point, we have changed the language in the revised manuscript to “single TSS driven” and “multiple TSS driven”.

11. For Pol II scanning, please add references to Giardina and Lis (10.1126/science.8342041), and Kuehner and Brow (10.1074/jbc.M601937200).

These early references to Pol II scanning have now been added. Our thanks to the reviewer for pointing these out.

12. PDF p. 12 "Candidate initiator motifs within 20 bp of the CA/ non-CA initiation site had slightly more initiation signal on the non-CA allele compared with the CA allele, consistent with the shooting gallery model but inconsistent with the prevailing model of independent TSNs expected in mammals (Figure 4B; purple). By contrast, TSNs where both alleles contained a CA dinucleotide (n = 8,147 [brain] and 10,113 [liver]) did not show this same effect (Figure 4B; gray)."If usage of YR elements in mouse examined, I think there is a hierarchy -> CA>TA{greater than or equal to}TG>CG. The analysis could be done differently to reflect this (CG being by far the most disfavored but also the most prevalent in TSS/TSR regions). What about consensus initiators (more information than just YR)- do they behave differently or the same? Is there a more sophisticated analysis that instead looks at fraction of distribution that is lost from max TSN vs fraction gained in next YR based on identity of that YR?

We explored the hierarchy of initiator dinucleotides in the revised manuscript. In our new analysis, we examined the magnitude of allelic difference in candidate initiator positions conditional on specific dinucleotide changes at max TSSs. Our results are largely consistent with the hierarchy expected by the reviewer: CA initiators which change to TA have the lowest magnitude of shift in initiation frequency between alleles, whereas CA dinucleotides that change to CG have the largest magnitude. Intriguing, CA to TG changes were intermediate between CA to TA and CA to CG. The number of examples for CA to TG is much lower because it required two DNA sequence changes. Therefore, we also examined TA to TG changes directly. This revealed that changes in initiator sequence from TA to TG was associated with a higher Pol II on the TA allele. Both of these results suggest a hierarchy where CA > TA > TG > CG.

We now describe this result in the revised manuscript and show the result in the revised Figure 2B. See especially the final paragraph of the section "Widespread genetic changes in transcription initiation":

“We used genetic differences between alleles to define the relative strength of different initiator dinucleotides. The initiator motif is perhaps the best characterized feature of Pol II initiation and is most commonly characterized by a CA dinucleotide, but the sequence preferences of the initiator motif are weak and other dinucleotides are relatively common (Smale and Baltimore, 1989). We examined how changes between initiator dinucleotides affected the abundance of ChRO-seq reads. We identified the set of all max TSSs which had DNA sequence differences between B6 and CAST alleles. Our analysis revealed a hierarchy of initiator dinucleotides that impact initiation frequency with different magnitudes (Figure 2B). CA initiators which changed to TA had the lowest magnitude of shift in initiation frequency, whereas CA dinucleotides that change to CG have the largest magnitude. CA to TG changes were intermediate between CA to TA and CA to CG. The number of examples for CA to TG was much lower than other dinucleotide combinations because it required two DNA sequence changes. Therefore, we also examined TA to TG changes directly. This revealed that changes in initiator sequence from TA to TG were associated with a higher Pol II on the TA allele. Our results suggest a hierarchy of initiation frequency, in which Pol II prefers to initiate at a CA dinucleotide, followed by TA, TG, and finally CG.”

We have looked into exploring more complicated consensus initiators as well, but unfortunately there are just not enough examples to explore all of the possible combinations rigorously.

Or simply whether the distribution in the TSS/TSR shifts upstream or downstream on average? With the sample size differences between affected and unaffected, would bootstrapping of the unaffected be a useful way to detect significant difference of the affected sample?

Our view is that our data is most consistent with a model in which the Pol II moves both upstream and downstream in roughly equal proportion. One important point to understand when interpreting Figure 4C is that there is an ascertainment bias in the discovery of tagged alternative initiator dinucleotides between upstream and downstream: because we are conditioning on a SNP at an initiator element in the center, every candidate TSS upstream of that SNP will be tagged. Downstream, however, requires a second SNP downstream of the tagged TSS. Since there are larger numbers of candidate alternative initiator elements upstream, the downstream box and whiskers plots are noisier. We think this explains why the [5,10] interval is not statistically different from the “*No SNP in Inr*” group. To clarify this point in the revised manuscript, we now include the ‘n’ underlying each window that have SNPs (purple, Figure 4C) box and whiskers plot above the plot.

Due to this ascertainment bias, we think the more relevant statistic is the magnitude of shift between the “*SNP in Inr*” test group and the “*No SNP in Inr*” control. The magnitude of difference is symmetric, with larger magnitudes, on average, close to the candidate TSS and declining magnitude further upstream or downstream (with the sole expectation of the [5,10] window, which could be attributed to a relatively low n=44 in that window). We think this supports our model in which changes in initiation frequency are symmetric when an Inr sequence is changed. This also supports the proposed brownian motion mechanism.

In text the caveat of multiple SNPs having differential effects on potentially overlapping PICs should be made stronger- for example affecting an upstream site through DPE etc.

We agree with this comment. We have updated the revised text to include this as a potential caveat of our analysis. See in particular paragraph 4 in the discussion, which reads:

“Another possible interpretation is that changes in the CA dinucleotide sequence alter DNA elements that support partially overlapping PICs (e.g., CA -> TA dinucleotide change makes the sequence slightly closer to a TATA box that could support initiation further downstream). Although we do not directly rule out this alternative interpretation, in our view it seems less likely because the effects we observe are constrained within a fairly narrow window, because the effects we observed are so consistent when conditioning on changes in the initiator DNA sequence motif, and because DNA sequence changes in the TATA box did not have as large of an impact on the initiation site. In either case, however, it is clear from our data that changes in the DNA sequence of initiator elements tend to increase the use of candidate initiators nearby.”

13. Figure 4C "The box plots show the distribution of ChRO-seq signals ratios at TSNs with any variation of the initiator motif (including CA, CG, TA, TG)"This sentence could be interpreted as "variation in the initiator motif" meaning SNPs in the initiator motif, when what is meant is all four YR sequences were considered.

We have updated the figure caption to read: “The box plots show the distribution of ChRO-seq signals ratios at TSSs with any YR dinucleotide (i.e., CA, CG, TA, TG) in both alleles”.

14. Figure 4D. The sequence relative to the background should be made clear. Potentially just label both B6/CAST positions that are different, especially because it is not clear that the B6 color code sequence actually lines up given how the genome browser is displaying adjacent positions.

We have added both B6 and CAST alleles that differ in the revised manuscript as a separate track that lines up with the ChRO-seq signal positions. We have also clarified that the B6 reference color codes line up with the ChRO-seq data in the revised legend (we note that ChRO-seq signal is slightly wider, but is aligned at the center of each SNP). Therefore, the CAST signal should be identical at positions which are not marked as SNPs.

15. PDF p. 16 "Changes in pause position were correlated with changes in the size of the insertion or deletion, such that the distance between the max TSN and the pause site in the native genome coordinates was typically less than 5 bp (Figure 5C)."Is it meant here that there appears to be a defined pause distance and therefore with INDELs pause site moves relative to genome, but stays consistent in number of bases between TSN and pause? Otherwise, I am having a hard time understanding what is meant here.

Yes, the reviewer understood correctly. We simply mean that the distance between the TSS and the pause position does not change very much (usually +/- 5bp). We have updated the text to use the language suggested by the reviewer, which we agree is clearer than what we wrote in our first manuscript draft. The updated sentence now reads:

“Changes in pause position were correlated with changes in the size of the insertion or deletion, such that the number of bases between the max TSS and the pause was typically less than 5 bp (Figure 5C).”

16. PDF p. 16 "Allelic differences at the C base had the strongest association with the pause, followed by the -2 and -3 G bases"This is unclear. It appears what is meant is that SNPs at 0, -2, -3 are most enriched for pauses sites with pause differences- referral to bases C,G is confusing because these are not present at all pauses, just enriched across pauses? Whatever is meant here, please make more clear.

We have changed our language to clarify the point raised by the reviewer. The revised sentence now reads:

“Allelic differences at the RNA polymerase active site (usually a C) had the strongest association with the pause, followed by SNPs at the -2 and -3 position relative to the pause (usually a G; Figure 5D; Supplementary Figure 5B).”

17. "These observations support a model in which DNA sequence changes that disfavor pausing result in Pol II slipping downstream to the next pause site for which DNA sequence is energetically favorable, but maintaining physical connections that may exist between paused Pol II and the PIC."It is unclear what is meant by "maintaining physical connections between the pause and the PIC"

We agree that the sentence was unclear, and we have removed that second portion. The revised sentence now reads:

“These observations support a model in which DNA sequence changes that disfavor pausing result in Pol II slipping downstream to the next pause site for which DNA sequence is energetically favorable.”

We meant to refer to the connections between paused Pol II and TFIID described by Fant *et al.* (2020) Mol Cell, which is supported in our study by the constraints on TSS-pause distance despite indels. We think these points are adequately made elsewhere in the Results and Discussion, and we agree that it was confusing as stated.

18. Figure 6A. What about any downstream changes associated with a AT window or adjacent to it? Are the AT events enriched inside larger strain-specific domains? How do these relate to other aspects of altered expression in region.

We performed two new analyses to address the link between AT windows and nearby gene expression programs.

– Do AT windows occur near other transcriptional changes? First, we asked whether AT windows are more likely to have adjacent genes that have an allelic bias compared with cases in which a gene does not have an AT window. We found that AT windows are more likely to be located adjacent to genes with allelic bias (liver: OR: 1.65, *p* = 1.83e-15; brain: OR: 1.23, *p* = 0.018; Fisher’s exact test). This result is robust to the most important ascertainment biases that we can think of: To avoid a bias in which AlleleHMM picks out a larger region, we performed the same test but using only transcription units on the opposite strand, which showed the same enrichment (liver: OR:1.33 , *p* = 1.32e-05; brain: OR:1.16 , *p* = 0.09 ; Fisher’s exact test). This enrichment was also not explained by confounding effects of gene expression, as there was no significant difference in expression of the nearby gene between these groups.

– Do strain affect domains that contain an AT window contain more transcription units? Second, we found that strain effect domains that contain an AT window tend to have larger numbers of transcription units inside of them (Two-sample KolmogorovSmirnov test, LV:p-value = 7.605e-14, BN: p-value = p-value = 0.01587).

We think these are potentially interesting observations and we have noted them in the revised manuscript. See especially paragraph 3 in the section entitled: “Allelic changes in gene length caused by genetic differences in Pol II termination”, which now reads:

“Next we examined how allelic termination windows were associated with changes in nearby transcription. Allelic termination windows were more likely to occur in highly expressed transcripts, which may partially reflect increased statistical power for detecting changes supported by larger numbers of reads. Additionally, allelic termination windows were more likely to have allelic changes in the transcription level of an adjacent transcript compared with matched transcripts without an allelic termination window (liver: odds ratio = 1.65, p = 1.83e-15; brain: odds ratio = 1.23, p = 0.018; Fisher’s exact test). We tested whether the association between allelic termination and adjacent transcript expression was also found when allelic termination and the adjacent transcript were encoded on opposite strands, hence avoiding the interpretation that AlleleHMM was more likely to detect allelic termination windows near an allelic difference with a large magnitude. The enrichments we observed were still significant in this more restrictive test (liver: odds ratio = 1.33, p = 1.32e-05; brain: odds ratio = 1.16 , p = 0.09; Fisher’s exact test). Intriguingly, the expression of nearby transcription units was frequently highest on the allele that terminated early (liver: odds ratio = 4.97, p = 2.2e-16; brain: odds ratio = 9.67, p = 2.2e-16; Fisher’s exact test). Taken together, these results suggest an association between the length of post-poly-A transcription and the expression of nearby transcripts.”

In the figure example there does look like there is a potential CAST leaning Liver dREG in the figure.

We clarified our meaning in the revised manuscript. Briefly, in order for dREG to explain allelic changes in Pol II density across the AT window, the dREG site has to be located at the beginning of the AT window. We did not find any dREG sites starting at the beginning of the AT window for the example shown in Figure 6A. We think the confusing point here is that because the gene in Figure 6A is on the minus strand, the start of the AT window is actually on the right-hand side of the region indicated. To clarify this point in the revised manuscript, we marked the start of the AT window by a triangle.

How does the height of the peaks on the "all" browser track relate to the Cast and B6, it does not look like sum of the two plus reads that can't be distinguished? Is it instead only reads that can't be distinguished? Is this different from what is show for TSN/TSS browser views?

We appreciate the reviewer catching this – we had an error in these tracks that prevented many CAST reads from being included in the figure. We have generated new browser tracks for the revised manuscript. In our revisions, the track labeled “ALL” represents all of the signal, not just the reads which cannot be labeled as “B6” or “CAST” (i.e., the same as in previous figures).

19. For considerations about association of energetic favorability of difference sequences, especially as relates to pausing, a critical caveat is any bias in sequence that favors detection by ChRO-seq, which will only be a subset of elongation complexes.

We agree with this point, which affects pretty much all sequencing studies. We have described potential sequence bias in detection, as well as the related issue of size bias from Illumina instruments, in the revised Results and Discussion sections.

Our intuition is that DNA sequence changes affecting the middle of an RNA insert (i.e., not the 5’ or 3’ base), especially those which constitute only a single base difference, are unlikely to have a large impact on detection in ChRO-seq. Several of our more novel results are supported by SNPs in the middle of the RNA insert, including the association between initiation and AT content downstream of the initiation site, and increased frequency of changes in G content inside of the paused Pol II active site.

In fact, we think there is a significant benefit to using an allele specific system in this type of analysis. Many studies (including many of our own) examine the effect of DNA sequence by comparing sequence content between different genes across the genome. This approach effectively assumes that enough sequence differences are present across the genome that technical factors will average out. However, as the reviewer points out, a statistical interaction between position in an insert and base content bias (e.g., GC-rich regions being slightly less likely to amplify, as observed in the early days of short read analysis) could impact results. We think this type of bias is less likely in allele specific analyses because we are comparing nearly the same sequence which differs by only a small number of single nucleotide differences.

Potentially more problematic for our study are DNA sequence changes affecting the 5’ or 3’ end of the RNA insert. In our study, changes affected by potential ligation biases on the 5’ or 3’ end of an insert (i.e., the “A” base in a “CA” initiator element, and the “C” base in the pause position) are all in-line with existing studies and our prior expectations. Notably, sequence bias has been discussed as a caveat by Tome, Tippens, and Lis (2018) Nature Genetics (among other authors). However, Tome et al. suggested that there was a bona-fide increase in the frequency of a C base at the active site in paused Pol II, because a C base was *not* enriched in the active site of Pol II located in the gene-body (i.e., the enrichment in C was specific to paused Pol II).

This argument seems fairly convincing to us.

To point out these potential limitations, we now write in the Discussion section:

“The use of an F1 hybrid system does, of course, have a number of limitations as well. Any DNA sequence variation between alleles that affects the probability of detection, including either DNA sequence variation or size bias in the RNA insert, could impact detection differently between alleles. One place where this might occur is ligation bias for SNPs on the 5’ or 3’ end of an RNA insert. However, we think the effect of such sequence bias is relatively small in our study. Results that might be affected, including the A base in the initiator element (at the 5’ end of the RNA insert), and the C base in the Pol II active site (at the 3’ end of the RNA insert), are supported by existing literature (Kaufmann and Smale, 1994; Kuehner and Brow, 2006; Smale and Baltimore, 1989; Tome et al., 2018). Major new results reported here generally reflect SNPs inside of the RNA insert, making differences in detection a less likely explanation.”

We have also included an additional caveat in the section which discusses the comparison between alternative pause sites and indels, which is the analysis most likely to be affected by Illumina size bias. This now reads:

“Changes in pause position were correlated with changes in the size of the insertion or deletion, such that the number of bases between the max TSS and the pause was typically less than 5 bp (Figure 5C). Although this result may be influenced by fragment length bias introduced during sequencing, it nevertheless provides additional support for a model in which paused Pol II is placed in part through physical constraints with the pre-initiation complex (Fant et al., 2020; Kwak et al., 2013).”

20. PDF p. 30 "Comparison of AT consent between alleles" -> typo

We have fixed this typo. Our thanks to the reviewer for reading carefully enough to pick this out!

21. PDF p. 31 "One has an allelic difference in mature mRNA and the other does not. Those with an allelic difference in mature mRNA were defined as having the RNA-seq AlleleHMM blocks between 10Kb upstream of the AT windows to the end of the AT windows."I suspect the majority of reader would read allelic difference in mRNA composition would relate to altered mRNA content, either APA or alternative splicing, but here it appears to relate to strain-specific expression of the RNA that shows AT, but another reading is that this analysis detects APA via differential expression over region where 3'UTRs are different. Please make this more clear- it also could easily be shown that over gene itself – where detectable – there are not allelic differences and therefore allelic differences in 3' UTR or identification of allele-specific splicing really suggest altered mRNA processing.

We examined both allelic changes in the mRNA primary sequence and the mRNA expression levels. We have made numerous changes in the revised manuscript to clarify when we are talking about each of these individual features of the mRNA. First, we now use the language “primary sequence” to refer to changes in the RNA sequence of the mRNA (i.e., either APA or splicing). We also show examples in which these changes are clearly changes in primary sequence, because they uniquely affect the expression level of a single exon while leaving the majority of the mRNA unaffected (see the revised Figure 6—figure supplement 2A).

We also conducted a separate analysis in which we asked whether changes in primary sequence are associated with changes in mRNA degradation rate. In this analysis, we use allelic differences in the ratio of ChRO-seq to mRNA-seq signal between alleles. We found that changes in the mRNA primary sequence are, in fact, associated with increased variation in the ratio of ChRO-seq to mRNA-seq. To clarify that we are examining changes in mRNA degradation rate (i.e., expression), we have moved this analysis to a separate paragraph and clarified the topic and lead-in sentences.

22. Supplemental Figure 2C. "at sites showing allelic differences in TSSs driven by a single base in the brain." base -> TSN to be more clear.

We changed “base” to “TSS” for clarity, as suggested by the reviewer.

Reviewer #3 (Recommendations for the authors):This manuscript from the Danko lab describes nascent RNA-sequencing of eight tissues collected from heterozygous F1 mice. Using the known sequences of each parental allele, nascent RNA was analyzed to define regions of strain effect, meaning there was a significant difference in sites of transcription initiation, transcriptional pausing, or termination depending on the strain. The authors then investigated genetic variation between parental alleles to shed light on specific nucleotides or sequence contexts of functional relevance. A large variety of different aspects of transcription are touched on, none in great depth, and the findings are generally in agreement with earlier work using alternative methods. Due to the low number of sites detected where differences between strains were apparent, the study is somewhat underpowered to make strong or new conclusions. In addition, it was not discussed whether small (5-10bp) shifts in sites of transcription initiation or pausing would have relevance for gene expression, making it unclear how to connect these findings to a biological effect.Strengths of this study include the clever study design, and sophisticated computational approach. This work could perhaps best be characterized as a methods paper, demonstrating a number of possible ways that F1 hybrid mice can be used to study gene regulation. In this regard, the authors succeed nicely.Weaknesses of the work derive from the small number of events detected wherein sequence differences between the parental strains appear to alter transcription initiation, pausing or termination. As a result, this work isn't able to tease out new mechanisms or bring us closer to being able to predict how sequence changes would affect transcription levels, but instead confirms or supports existing models.

We appreciate the reviewer’s comments and constructive review. We agree with the main points raised here. We do think there are several places where our manuscript significantly advances existing knowledge (models of initiation, for instance), but we agree that these more interesting findings were buried and understated in our original draft. We have endeavored to highlight the places where our revised manuscript advances existing knowledge, as described in detail below.

Results concerning the mechanism of search by RNA polymerase II for a site to initiate have more promise to reveal new findings. It would be interesting to provide compelling evidence of Pol II scanning in mammals, as it is known to do in yeast. The analysis as presented seems underpowered by a small number of instances. Perhaps the authors could dig into their data for other tissues or aggregate across tissues to extract more from this analysis?

The revised manuscript digs into the Pol II initiation models in much more depth. We do believe that we present evidence of brownian motion of Pol II along the DNA during initiation; we also think this scanning is quite distinct from the initiation process in yeast. We have made extensive changes to the text and figures that clarify what we have learned about transcription initiation, and supported these findings by new analyses. Finally, to lead readers through our findings more effectively, we have added figures that illustrate the initiation models in mammals and how our work differs from existing proposals that are in the literature.

Finally, results are presented indicating that differences in locations of transcription termination can be detected between the strains. This is the one instance where the authors probe steady state RNA-seq to determine if such differences affect levels of mature mRNA. In this case, the answer appears to be generally 'no', suggesting that the cleavage and polyadenylation site is the same between strains, and it is only the location of Pol II termination after mRNA is cleaved and released that differs between alleles. Thus, this finding, as interesting as it may be, does not appear to have a biological consequence.

We have made substantial changes to the section on Pol II termination to clarify the text and add additional analyses.

We have conducted a new analysis to support this section which asks whether changes in the position of transcription termination are enriched for changes in mRNA primary structure. We found a 2-4-fold increase in allelic differences in mRNA inside of allelic termination windows relative to transcription units with no evidence of allelic termination. We show examples in which these changes reflect differential inclusion of final exons between alleles. These results support that while changes in transcription termination frequently do not alter mRNA primary structure, they do in a significant fraction of the cases.

Additionally, we have also examined whether changes in mRNA primary structure affect degradation rates. The reviewer correctly states that our original manuscript did not find a significant association. However, during our revisions, we found and corrected a programming bug affecting the analysis comparing changes in mRNA degradation rates. After correcting this error, mRNAs with changes in mRNA primary structure (20-40% of the changes in termination) are in fact enriched for changes in mRNA degradation rate. Thus, we have, at least in this case, demonstrated a biological consequence on both primary structure and mRNA degradation rate.

These results are presented in the revised Figure 6, the associated supplementary figures, and in the Results sections.

Overall:I was impressed with the approach but underwhelmed with the resulting findings.I find the analyses to be lacking in depth and presented more as a hodge-podge of vignettes rather than a connected story. As a result, there are some potentially interesting nuggets of information uncovered, but these are often presented without biological context or interpretation. As a result, I wasn't sure what the 'take-home' messages of the work might be.For example, results concerning transcription initiation are solid, but unsurprising: mutations in the core of the Initiator motif (e.g., the A residue with the highest information content and weight in a PWM) are the most important for determining levels of initiation at that location, and the AT-richness of the transcription bubble and energetics of melting play some role in choice of initiation site. Nice findings, and rigorous work, but the authors might need to collect more data or dig deeper into the analyses to break new ground.

As noted above, we agree with the reviewer that our presentation was underwhelming in the original manuscript. In response, we have made extensive changes to the revised manuscript which are intended to focus on a connected story: How genetic changes affect the RNA Pol II transcription cycle. These changes are extensive, and include: (1) Removing the section and most of the content on genomic imprinting, which, while interesting, was a departure from our primary message; (2) Adding new computational analyses that support models of brownian motion, transcription initiation, and pause; and (3) Adding new analysis and re-writing to clarify our results on the connection between transcription termination, alternative polyadenylation cleavage, and the effects of these genetic differences on mRNA stability.

Specific concerns are:1) the authors state that strain-effect domains that showed allelic imbalance 'were generally composed of multiple transcription units.' however, figure 1E shows that >40% of the strain effect domains contain only 1 annotated gene, and ~30% contain 2 genes. If the figure is accurate, and >70% of the strain effect domains contain 1-2 genes, I don't see how this is in agreement with the text. The percentage of imprinted domains with >1 gene seems higher, but with only 28 domains to study, it is unlikely that one can draw strong conclusions from this.

The reviewer is correct that changes affecting 1-2 transcription units are most common for starin affect domains. These comprise 42% of domains identified in our analysis. We have taken several steps to refine our revised manuscript to avoid confusion and increase the impact of this section. First, we wish to clarify that the histograms in the original manuscript were based on bins of 2 genes in each bin. We have changed the histogram in the revised manuscript to use 1 gene bins, and to show the number of domains with 0 genes. Second, we have taken more care with our language in the revised manuscript. Our revised Results section now reads that domains were “frequently composed of two or more transcription units”, which we think matches the result accurately. Third, we have significantly changed the writing to lead into this result in a more biologically motivated manner that we think will resonate much better with readers.

Our intention in this section was to show that genetic variation between alles frequently hits locus control regions or other regulatory processes affecting multiple genes, rather than regulatory regions impacting individual genes. To explore this result in more depth than was included in our original manuscript, we have added new analyses that ask whether changes in adjacent genes occur more frequently than expected by chance (we found strong evidence that this is indeed the case). Our new analysis is presented in the revised Results section (see paragraph 3 in the section titled: “Atlas of allele specific transcription in F1 hybrid murine organs”):

“Examination of genome browser tracks showed that nearby genes frequently showed similar patterns of allelic bias. For instance, the imprinted domain associated with Angelman syndrome consisted of twenty ncRNAs and four genes that were consistently transcribed more highly from the paternal allele and two genes transcribed from the maternal allele in brain (Figure 1D). To test whether genes generally showed more positional dependence than expected by chance, we asked whether genes having evidence of allelic bias tended to cluster. Indeed, we found that genes with allelic bias were significantly more likely to have allelic changes in the expression of an adjacent transcript compared with genes which were not changed (brain: p-value = 5.35e-4, liver: p-value < 2.2e-16; Fisher's Exact Test).”

2) I will note that I was struck by how few imprinted domains were observed. Do the authors think that this results from the short read length of Chroseq, which makes it rare to have a strain specific SNP in a read? If so, have the authors considered using RNA-seq to define differences in RNA levels between strains, and then to use Chroseq to investigate what steps in the transcription cycle and which SNPs might underlie these effects? Naively, it seems like including a method like RNA-seq with longer reads and the ability to determine whether changes are biologically meaningful could strengthen the work and allow more interesting or novel conclusions to be drawn.

We appreciate this comment and these suggestions. In order to better focus our revised manuscript, we have removed the sections on genomic imprinting. We are in agreement with all reviewers that these sections, while interesting, largely do not advance much beyond previous literature and distract readers from the main point we would like to make in our paper: namely how DNA sequence changes affect the various stages of Pol II transcription.

Having said this, we do wish to discuss the reviewer’s (correct!) perception that the number of genes identified as imprinted in our study is small. We think that our statistical power based on ChRO-seq data alone is quite high for individual genes and for large domains, especially when analyzed using AlleleHMM (see (Chou and Danko; Nucleic Acids Res. 2019) for a comprehensive analysis of statistical power). We think that the small number of genes results from two factors, which are consistent with previous literature: First, most studies on this topic (with a few notable exceptions) find that the number of imprinted genes in mammals is on the order of low-hundreds (see, for instance, a recent review by Kobayashi (2021) Front. Cell Dev. Biol. which places the number at ~260 in mice). Second, most of the imprinted genes have allele specific expression in developmental tissues, including placenta, with relatively smaller effect sizes and numbers of genes in adult organs that we analyze here (see, in particular, Figure 4 of Andergassen et al. 2017, *eLife*: https://elifesciences.org/articles/25125). Thus, between these two considerations, we think that our results of 28 imprinted domains (each of which contains multiple genes) and our genelevel analysis presented in the revised Figure 1C that conservatively identifies 51 candidate annotated genes, are of an order of magnitude that is consistent with existing literature.

3) Presentation of some results could be improved to increase clarity. In particular, I was interested in the data shown in Figure 4, but struggled for some time to truly understand exactly what was being shown. I worry that the presentation style will limit the readership.

We agree, and we have made several changes in the presentation of Figure 4 to make it more accessible.

– First, we have expanded the models depicted in the figure to separately show current initiation models in yeast and mammals, proposed by the Kaplan and Price labs. These models illustrate the expected change in the distribution of transcription initiation given a SNP in a strong initiator dinucleotide. We have also added a third model to illustrate the brownian motion idea that we propose is a better fit to our observations than either of the existing models.

– Second, we significantly expanded writing in the Results section to explain each model and the expected outcome in more detail.

– Third, we have added new analysis to explore alternative interpretations of our results. Collectively, these revised analyses continue to support the model of initiation depicted in the revised Figure 4E. This analysis is described in the final paragraph in the section “Models of stochastic search during transcription initiation”.

We think these revisions have helped to strengthen our model and improve the quality of the explanation to readers from many disciplines.

[Editors’ note: what follows is the authors’ response to the second round of review.]

Reviewer #1 (Recommendations for the authors):This new version of the manuscript has addressed many of the issues raised by the referees on the original version. It is more focused and concise. The addition of new data has raised several issues that need to be addressed.The authors have added new data concerning how SNPs affect Pol II transcription termination. The data show that many SNPs lead to longer primary transcripts raising the possibility of use of alternative polyadenylation sites. To address this, the authors sequence the corresponding PolyA+ transcripts and claim that a significant fraction show an altered primary structure, in some cases with inclusion of additional exons. This section is very confusing as the authors make no mention of use of alternative polyadenylation sites. They are not mapped or discussed.

We have re-written sections in which we analyzed mRNA-seq data to understand the impact of allelic differences in termination. We think the revised text is much less confusing. The revised paragraph is provided here for the convenience of the reviewer:

“To determine whether allelic differences in termination influence the primary structure (i.e., the processed mRNA sequence) of the mature mRNA, we next sequenced poly-A enriched mRNA from two liver and brain samples. Full-length mRNA-seq data can detect allelic differences in the frequency of use between exons within a gene based on the relative mRNA-seq signal in each exon. Differences occurring in exons at the 3’ end of the gene are likely to reflect alternative use of polyadenylation cleavage sites between transcripts, a mechanism which has recently been reported to be subjected to genetic changes in humans (Mittleman et al., 2021, 2020). For instance, we identified an unannotated 3’ exon in Sh3rf3 that most likely reflects the use of a novel polyadenylation cleavage site unique to the allele having a longer transcription unit (Figure 6—figure supplement 2A). In most cases, including Sh3rf3, RNA-seq signal in the novel candidate exon was lower than in the annotated 3’ exon, indicating that novel exons were included in only a portion of the mRNAs produced from the allele with the longer transcription unit.

To map allelic changes to mRNA primary structure involving unannotated candidate exons, we used AlleleHMM to identify differential mRNA abundance between alleles using the mRNA-seq data. Our analysis determined that 21-41% of transcription units with differences in allelic termination had evidence of changes in mRNA primary structure identified using AlleleHMM (Figure 6D), an enrichment of 2-4-fold relative to transcription units with no evidence of allelic termination (p < 2.2e-16; Fisher’s exact test in both brain and liver). We conclude that changes in allelic termination were often associated with corresponding changes in the primary structure of the mature mRNA.”

Also, it is not clear why having a longer primary transcript would alter the use of the original polyadenylation site. While 21-41% of transcription units have altered primary mRNA structure, does this mean that the major polyadenylation site is no longer used or used at a lower frequency. Which % of the total transcripts from a given gene actually show the altered primary structure. This issue also is neither mentioned nor discussed.

We fully agree with the reviewer that a longer transcript will often not affect the shorter polyadenylation cleavage site. Indeed, in cases that appear consistent with a new polyadenylation cleavage site in the longer transcript (e.g, *Sh3rf3* in Figure 6 —figure supplement 2A), we have noticed that candidate alternative 3’ exons typically have a lower read depth compared with the annotated 3’ exon. We interpret this lower read depth to reflect a mixture of isoforms being expressed for the allele with a longer transcription unit, which includes both the annotated and candidate novel polyadenylation cleavage sites. Judging by differences in read depth, we think the fraction of total transcripts which show the altered primary structure are generally not a large fraction of the total mRNA. However, we caution that full length mRNA-seq is not the most direct strategy for measuring the frequency of alternative polyadenylation cleavage site use (a more direct strategy is a 3’ end enriched RNA-seq protocol, such as those used by Mittleman et al. (Mittleman et al., 2021, 2020)). For this reason, we have thus far avoided estimating the percentage of each isoform.

To address this point, we have clarified in the revised manuscript that not all of the transcripts inside of a cell are altered by noting the difference in RNA-seq signal between annotated and novel exons. In addition, we have emphasized that novel isoforms likely reflect a minor portion of the total mRNA, which could explain why differences in mRNA stability occur only at a relatively small portion of genes with alternative termination sites.

Changes include several additions to the Results section, including:

“For instance, we identified an unannotated 3’ exon in Sh3rf3 that most likely reflects the use of a novel polyadenylation cleavage site unique to the allele having a longer transcription unit (Figure 6—figure supplement 2A). In most cases, including Sh3rf3, RNA-seq signal in the novel candidate exon was lower than in the annotated 3’ exon, indicating that novel exons were included in only a portion of the mRNAs produced from the allele with the longer transcription unit.”

And:

“We emphasize that novel isoforms are likely to be a minor portion of the mRNA produced from the allele with the longer transcription unit, which could help explain why differences in mRNA stability occur at a relatively small portion of mRNAs with changes in primary structure.”

And in the discussion, where we now write:

“We note that in cases where the inclusion of a novel exon appeared to be altered between two alleles, the novel isoform typically represented only a small portion of the total mRNA population, as judged by comparing the signal in the alternative and annotated 3’ exons. Nevertheless, differences in mRNA stability between alleles were more common in cases where alternative termination altered mRNA primary structure. These findings suggest that changes in Pol II termination can impact mRNA primary structure and ultimately mRNA stability.”

Reviewer #2 (Recommendations for the authors):The manuscript is improved and more focused. My concerns mostly relate to accessibility of results. Throughout the results the manuscript could be improved if framework for understanding results were more clear. Explicitly stating possibilities in each case and then indicate what types of changes the analyses detect and what they suggest.Abstract1. "Our results implicate the strength of base pairing between A-T or G-C dinucleotides as key determinants to the position of Pol II initiation and pause."This is maybe a little strong for the abstract.

We have changed this sentence in the abstract to be a more genetic identification of both known and novel DNA sequence motifs.

Results2. "Our results suggest a hierarchy of initiation frequency, in which Pol II prefers to initiate at a CA dinucleotide, followed by TA, TG, and finally CG"I believe this also fits the observed usage hierarchy observed in TSS-seq in humans/mice (potentially check if there are appropriate references for this but note that usage should be normalized to dinucleotide frequency). Also note, it is different than yeast preference https://doi.org/10.1101/2021.11.09.467992

We have indicated that the mammalian hierarchy differs from yeast in the revised manuscript.

We agree with the reviewer that the CA > TA > TG > CG hierarchy we observed is largely consistent with previous mammalian studies. For instance, according to Javahery et al. (1994) and Ngoc et al. (2017), the A at the +1 position is *extremely* important; initiation is often lost when this A is changed to another base, even to a C (see also the excellent review by Smale and Kadonaga (2003)). Likewise, we think the prominent G in the +1 position in motif logos by Carninci et al. Nat Gen (2006) (see Figure 2 B-E) mostly likely reflects the propensity for initiation from CpG islands in mammals, and hence is not properly normalized by dinucleotide frequency as required for a direct comparison with our work. With this said, we did not find a definition of the hierarchy in mammals that is comparably high-throughput to the one indicated by the reviewer in yeast (by Zhu et al. bioRxiv (2021)), making a direct comparison with our work more challenging.

We now note these parallels with previous literature in the revised manuscript, writing: “This hierarchy observed in mammals differs substantially from yeast, which favors CG over either TA or TG (Zhu et al., 2021), but is more closely aligned with the importance of A in mammals based on promoter mutagenesis studies (Javahery et al., 1994; Ngoc et al., 2017; Smale and Kadonaga, 2003).”

3. "We therefore developed a statistical approach based on the Kolmogorov-Smirnov test to identify differences in the shape of the distribution of Pol II initiation (see Methods)."This potentially seems fine but note that there are other definitions of shape and tools used to determine differences in index. I suspect the original "shape" index might not necessarily work for more subtle changes (see https://genome.cshlp.org/content/21/2/182.long) but that might be what readers might think shape of TSS distribution is referring to. For potentially alternative framework or useful statistical approach to examining differences in distributions see: https://academic.oup.com/nargab/article/3/2/lqab051/6290625

The revised manuscript articulates the definition of “shape” used in the present study more clearly as “any allelic difference in the TSS distribution within a TSC”. We also now cite the Policastro et al. (2021) paper, which we think is fairly close to what we hoped to achieve with the KS test (indeed, it looks like Policastro used a KS test as one of the strategies in their package). We still think that “shape” is the most intuitive short-hand word to describe shifts in TSS usage identified by the KS test (though we are happy to revisit this if the reviewer has specific suggestions).

As an aside, we like the earth mover’s distance metric introduced in the Policastro et al. study and have adopted ideas from this paper for a related project that is still in progress.

4. Section starting with sentence "Next we examined the distribution of SNPs centered on allele specific max TSSs"I think these paragraphs could be more clear if possibilities for differences were imagined for the reader and then the analyses introduced. What is being done here is asking where SNPs are relative to the affected site when the change between alleles appears to be driven by differences in a single site or multiple sites. And the answer is the SNPs tend to be right on top of the affected TSS.

We have added a new topic sentence that improves clarity by laying out the positioning of potential DNA sequence elements relative to the TSS.

The revised sentence reads: “Although the PIC occupies DNA spanning -30bp to +30 bp relative to the TSS (Schilbach et al. 2017; He et al. 2016; Chen et al. 2021), transcription is also controlled by transcription factors that frequently have factor-dependent positional preferences within the nucleosome free region, located approximately -1 to -110 bp relative to the TSS (Core et al. 2014; Scruggs et al. 2015; Grossman et al. 2018; Tippens et al. 2020).“

5. Section just below above, paragraph starting with "Intriguingly, although the increased density of SNPs observed was largest near the Inr element, SNPs were enriched throughout the region occupied by the PIC in both single and multiple TSS driven allele specific TSCs (Figure 3C,E)."This section is relatively dense and not really accessible. This section appears to be asking whether SNPs might be affecting allele-specific expression through changes in and around TATA. The answer appears to be yes, to some extent. I think the paragraph could be much more clear in that regard. [Please also review the section on pausing to ask how accessible it is]

We have significantly revised the introductory sentences in this section in an effort to clarify our primary goal of asking whether SNPs in a TATA box correlate with allelic changes in initiation shape. The revised text now reads:

“We asked whether the TATA box, a canonical core promoter motif that binds TFIID, was also enriched for DNA sequence changes associated with allelic variation in TSS position. As the TATA box only occurs at ~10% of mammalian promoters (Carninci et al., 2006; Lenhard et al., 2012), we conditioned on the presence of a clear TATA-like motif on at least one of the alleles and asked whether SNPs affecting the TATA box correlated with the magnitude of effect on initiation. We first used a TATA motif that had a general enrichment for AT content, consistent with the degenerate nature of the TATA box in mammals (see Methods). We found that SNPs which changed the TATA motif had a positive correlation with allele specificity, with a slightly stronger correlation in brain than in liver (Pearson’s R = 0.09 [liver], n = 201; R = 0.18 [brain], n = 121). The positive correlation was marginally significant in the brain (p = 0.04), but not in the liver (p = 0.2). Similar results were also obtained with an additional TATA motif that was a stronger match to the classical TATA consensus (TATAAA; p = 0.078 [liver], n = 218; p = 0.051 [brian], n = 37). These results are consistent with core promoter motifs playing an important role in the position and magnitude of transcription initiation, but they appear in our analysis to be weaker determinants of the precise initiation site than the initiator motif.”

6. Section starting "Models of stochastic search during transcription initiation". If term "shooting gallery" is used to describe initiation by promoter scanning, it should be explained briefly ie. "a directional process where rate of catalysis and processivity and rate of DNA translocation determine distribution of TSS usage" and stressing that upstream TSSs will by default have priority over downstream ones though still reflecting innate Inr strengths.

We have explained the term “shooting gallery” in this section and emphasized the dependence on both the rate of DNA translocation and the strength of the initiator element.

The revised section now reads:

“Next we examined how SNPs that affect a particular TSS impact initiation within the rest of the TSC. In the prevailing model of transcription initiation in *S. cerevisiae*, called the “shooting gallery model”, after DNA is melted, Pol II scans by forward translocation until it identifies a position that is energetically favorable for transcription initiation (Braberg et al., 2013; Giardina and Lis, 1993; Kaplan et al., 2012; Kuehner and Brow, 2006; Qiu et al., 2020) (Figure 4A, left). Under the shooting gallery model, the transcription start site is determined by the rate of DNA translocation, resulting in a preference for more upstream TSSs, while still retaining the strength of initiator elements. In mammals, Pol II is not believed to scan, but rather each TSS is believed to be controlled by a separate PIC (Luse et al., 2020) (Figure 4A, right). We considered how mutations in a strong initiator dinucleotide (CA) would affect transcription initiation under each model. Under the yeast model, we expected CA mutations to shift initiation to the next valid initiator element downstream.

Under the mammalian model, we expected each TSS to be independent and therefore a mutation in the TSS would have no effect on the pattern of nearby initiation sites.”

7. "This indicates that changes in allelic termination often had a corresponding impact on the primary structure of the mature mRNA.""impact" seems to suggest causality so potentially use a different word? I think you are just saying there is an apparent correlation when IDing change in termination with IDing altered primary structure (though potentially should also do the analysis in reverse, if possible).

We agree with this point. We have modified the text to avoid any claims of a direct causal mechanism. The new text reads: “This indicates that changes in allelic termination were often associated with corresponding changes in the primary structure of the mature mRNA.”

Figures8. Please check all figures and legends. PDF conversion appears to have corrupted spacing on many graph legends, but additionally, there are a few "muntants" and a "browian".

Our thanks to the reviewer for catching these mistakes! These errors have been corrected. We have checked the figures and legends carefully and corrected a few other PDF rendering issues.

9. Figure 2-sup 1"(B) Scatter plot shows the number of TSSs in a TSC as a function of the read counts in theTSC."Consider analyses but only counting sites with some threshold of total usage, e.g. 1% or 2%- this might give a better idea about real vs just seeing something at a position because read depth increases.

Our primary use of this figure panel was to reproduce the general features observed by Tome Tippens and Lis, Nat Gen (2018) (see Figure 1D in their paper). We cite this panel as an additional sanity check that paired-end ChRO-seq data recovers TSS clusters with similar properties to those obtained using coPRO protocols, which involve enzymatic cap selection. We do agree that the reviewer’s suggested changes would be useful, and we plan to explore them in the future as new projects focus on resolving the properties of TSCs.

Figure 6-sup 1 A and B legends are switched.

We have corrected this mistake. Our thanks to the reviewer for catching this!

Reviewer #3 (Recommendations for the authors):In this revised submission, the authors have addressed the major concerns of previous reviewers and, as a result, produced a much revised but stronger and more focused manuscript. The manuscript covers an important topic, the genetic determinants of RNA polymerase II activity using an excellent and well designed system, the F1 mouse. Below I outline my one concern which should still be addressed.Much of their work on the shape of transcription initiation hinges on the Kolmogorov-Smirnov test, which -- while statistically valid, is quite permissive in calling two things different. Given the high number of initiation regions with KS-test differences, the fact that many of these do not overlap strain effect domains, and the fact that replicates in nascent protocols can have subtle shape differences by random chance -- I have to wonder how many false calls they would expect using this KS approach. As they condition on B6 vs CAST alleles for the KS-test, one possibility would be to apply the test within strain but across replicates to estimate a false positive rate -- though admittedly data amount may be limiting in this scenario.

We agree that the Kolmogorov-Smirnov (K-S) test can often be too sensitive in genome-wide applications and could potentially introduce false positives into analyses of both initiation and pause. In response, we conducted the analysis comparing biological replicates suggested by the reviewer. We focused on replicates F5 and F6, which contributed heart, kidney, and skeletal muscle tissue used to analyze the position of paused Pol II. We applied the K-S test between biological replicates in a manner that approximates its use between alleles as closely as possible: using only reads tagged with a SNP and using the same filters (at least 5 reads on each allele, within a dREG site, etc.). We note that both our heuristics to filter the universe of sites tested and our use of the K-S test were nearly identical when applied for the discovery of allelic differences in initiation and pause shape, and therefore we believe that the empirical evaluation of the K-S test noted below provides accurate information about the empirical false discovery rate for both applications.

The application of the K-S test did not identify nearly as many candidate pause differences between biological replicates as it did between genetically distinct alleles: Between replicates, the K-S test identified n = 35, 2, and 220 candidate differences at a 10% FDR in a universe of 8985, 7964, and 8204 potential candidate pause regions in heart, kidney and skeletal muscle, respectively (representing 0.02 to 2.5% of sites affected). Conversely, between B6 and CAST alleles, the same test identified 1202, 822, and 927 candidate differences at a 10% FDR in a universe of 7748, 7178, and 7843 pause regions (representing 12-16% of the universe of sites).

When interpreting these differences, it is important to note that there are potentially numerous sources of confounding biological and technical variation between different biological replicates. Although we did our best to control for these (for example, by harvesting mice at the same time of day, controlling mouse age, quickly placing organs on ice, minimizing RNA degradation, etc.), it is nevertheless possible that there are some bona-fide differences in either the trans environment or technical batch effects between biological replicates. We note that these same biological and technical factors would not confound the better-controlled test between two alleles within the same tissue sample, as the set of technical and biological factors in the trans environment are identical between the B6 and CAST alleles within the same nucleus.

We think this test supports the idea that the majority of differences detected using the K-S test reflect genetic differences between alleles. This test complements the other sanity-checks and filters we implemented in our previous work (e.g., requiring a minimum read density, use of an F1 hybrid design, extensive checking of specific examples in a genome-browser to ensure they look reasonable, etc.). Therefore, we have noted this test as an additional sanity check in the revised Methods section.

Finally, we also wish to note that it is not too surprising to us that there are many initiation and pause sites that undergo small changes in shape based on genetic factors which are not also identified in larger strain-effect domains. We think it is reasonable to speculate that there is not a strong selective advantage for the organism to preserve a specific initiation site rather than an alternative a few base pairs apart. Combined with weak DNA sequence information encoding the precise transcription start site, this system may be prone to subtle differences in the precise initiation site used across a population. These observations, and the speculation provided here for the benefit of the reviewer, may be in-line with those recently pre-printed by the Andersson lab (Einarsson et al. 2021). We now cite this manuscript and its main idea that small differences in initiation sites can confer robustness to genetic perturbations in the revised manuscript.